# RAF inhibitors promote RAS-RAF interaction by allosterically disrupting RAF autoinhibition

Ting Jin[1], Hugo Lavoie[1], Malha Sahmi[1], Maud David[1], Christine Hilt[2], Amy Hammell[2] & Marc Therrien[1,3]

First-generation RAF inhibitors paradoxically induce ERK signaling in normal and tumor cells exhibiting RAS activity. Compound-induced RAF dimerization through stabilization of the RAF ON/active state by inhibitors has emerged as a critical contributing factor. RAF inhibitors also enhance RAS–RAF association. Although this event is thought to play a key role in priming RAF activation, the underlying mechanism is not known. Here we report that RAF inhibitors induce the disruption of intramolecular interactions between the kinase domain and its N-terminal regulatory region independently of RAS activity. This provides a molecular basis to explain the induction of RAS–RAF association by RAF inhibitors, as well as the co-operativity observed between RAS activity and RAF kinase inhibitors in driving RAF activation. Profiling of second-generation RAF inhibitors confirmed their improved mode of action, but also revealed liabilities that allowed us to discern two properties of an ideal RAF inhibitor: high-binding affinity to all RAF paralogs and maintenance of the OFF/autoinhibited state of the enzyme.

[1] Institute for Research in Immunology and Cancer, Laboratory of Intracellular Signaling, Université de Montréal, C.P. 6128, Succursale Centre-Ville, Montréal, QC, Canada H3C 3J7. [2] Oncology Discovery, Bristol-Myers Squibb Research, Princeton, NJ, USA. [3] Département de pathologie et biologie cellulaire, Université de Montréal, Outremont, QC, Canada H2V 1E8. Ting Jin and Hugo Lavoie contributed equally to this work. Correspondence and requests for materials should be addressed to M.T. (email: marc.therrien@umontreal.ca)

RAS–ERK signaling is generally initiated at the plasma membrane where ligand-bound receptor tyrosine kinases (RTKs) transduce signals to RAS GTPases by stimulating their GTP loading[1,2]. Downstream of RAS, mammalian cells express three RAF paralogs (ARAF, BRAF, and CRAF) that share a conserved C-terminal kinase domain (KD)[1,3]. They also comprise an N-terminal regulatory region (NTR) consisting of a RAS-binding domain (RBD), a cysteine-rich domain (CRD), and a Ser/Thr-rich region. The mammalian RAF family also comprises two KSR isoforms that share significant sequence identity with RAF proteins in their C-terminal kinase domain and present a related NTR organization with the exception that they contain a signature domain, called the coiled coil-sterile α motif (CC-SAM) domain, and lack an RBD domain[1]. In unstimulated cells, RAF proteins are sequestered in the cytoplasm as monomers in an autoinhibited state through an intramolecular interaction between their NTR and kinase domain (referred to hereafter as the RAF OFF-state). Upon RTK activation, GTP-bound RAS binds to the RAF RBD, which is thought to release the NTR–KD interaction[4–6]. This event is accompanied by dephosphorylation of inhibitory sites and phosphorylation of activating residues that respectively contribute to membrane anchoring and kinase domain activation[1]. Concomitantly, RAF proteins undergo kinase domain side-to-side dimerization[7]. This step allosterically drives catalytic switching to the ON-state and is essential for kinase domain activation. Here we refer to the ON-state as dimeric full-length RAF in which NTR autoinhibition has been relieved. Activated RAF proteins convey signals by initiating a phosphorylation cascade from RAF to MEK and MEK to ERK, which culminates in the phosphorylation of an array of substrates eliciting cell-specific responses.

Given the strong association of RAF activity with cancer, the past decade has seen the development of a broad set of ATP-competitive RAF inhibitors[8]. Some of these first-generation RAF inhibitors have shown impressive efficacy against metastatic melanomas harboring the recurrent BRAF[V600E] allele and have been approved for treating this patient population[9,10]. The clinical responses against BRAF[V600E]-dependent melanomas result from potent ATP-competitive inhibition of the monomeric form of this specific BRAF-mutant protein[11]. Unfortunately, acquired resistance to these agents invariably develops in part by mechanisms that stimulate RAF dimerization including upregulation of RTK signaling, RAS mutations, and BRAF[V600E] amplification or truncation[12–15]. Concurrently, tumors exhibiting RAS activity—owing to activating RAS mutations or elevated RTK signaling, but which are otherwise wild-type for BRAF—show primary resistance to RAF inhibitors[16–19]. RAF inhibitors were indeed found to induce ERK signaling in conditions where RAS activity is elevated and therefore enhanced tumor cell proliferation[16,17]. This counterintuitive phenomenon, known as the paradoxical effect, was also observed in normal tissues relying on physiological RAS activity and is the basis for some of the adverse effects seen with RAF inhibitors in melanoma patients[8]. The underlying mechanism results in part from the compound ability to promote kinase domain dimerization[16–18]. This event is not restricted to BRAF, but also involves other RAF family members and is dictated by the compound binding mode and affinity[16,18,20]. In brief, inhibitor-bound RAF kinase domains undergo a conformational transition to the ON-state enabling them to dimerize with, and allosterically transactivate in a RAS-dependent manner, RAF proteins unbound by the compounds, hence leading to downstream ERK signaling. We will refer to this class of compounds as ON-state inhibitors throughout the manuscript. Signal transmission is dose-dependent and thus inhibited when both protomers of a dimer are occupied by the compound. However, several chemical series appear to induce negative co-operativity within dimers in which compound binding to one protomer reduces the affinity of the compound to the opposite protomer[11,21]. Consequently, significantly higher drug concentration is required for inhibiting RAS-induced RAF dimers compared with BRAF[V600E] monomers. The structural basis for this is currently not fully understood.

Two strategies have recently been pursued to circumvent the limitation of first-generation RAF inhibitors. The first one focused on molecules with higher binding potencies to all RAF paralogs in order to saturate RAF proteins at lower drug concentration thereby minimizing paradoxical pathway induction. This led to a diverse set of so-called pan-RAF inhibitors. These molecules demonstrate promising activity in constitutively-activated RAS-mutant cells and animal models[22–25]. However, they generally exhibit strong RAF dimerization induction and thus still present some degree of paradoxical ERK signaling at sub-saturating doses, whose liability remains to be assessed in clinical trials. The second strategy consisted in designing compounds that do not paradoxically induce ERK signaling. This has given rise to "Paradox Breaker" (PB) molecules like PLX8394 that are derivatives of PLX4032/vemurafenib[26]. These molecules retained high potency against BRAF[V600E] and therefore should prove useful for treating BRAF[V600E]-dependent melanomas. However, while they did not induce ERK signaling in the few RAS-mutant cell lines that have been tested, they did not efficiently suppress it either and are thus ineffective against RAS-mutant tumors.

In addition to inducing RAF kinase domain dimerization, RAF inhibitors enhance RAS–RAF interaction[16,21,27,28]. Although the mechanism is not known, it suggests that binding events in the RAF catalytic cleft are allosterically communicated to remote RAS-binding surfaces enabling increased association with RAS. Understanding how inhibitors enhance RAS–RAF interaction may lead to the identification of better molecules for treating RAS-dependent tumors.

Here, we discovered that RAF inhibitors that promote RAS–RAF association also disrupt the intramolecular NTR–KD interaction taking place in BRAF and CRAF. Interestingly, release of this inhibitory interaction occurs independently of RAS binding to the RAF RBD. Moreover, while this event correlates with the compound potency to induce kinase domain dimerization, dimerization per se does not appear to be the sole factor. Indeed, the compound binding mode also emerges as an important contributing element. These findings provide a molecular framework to understand the co-operativity that exists between RAS activity and RAF inhibitors in driving paradoxical RAF activation. Finally, we profiled second-generation RAF inhibitors and show that pan-RAF inhibitors maintained the characteristics of first-generation compounds and effectively disrupted the NTR-KD interaction. In contrast, paradox breakers, which marginally induce dimer formation, minimally perturbed RAF intramolecular interactions and thereby have a considerably reduced propensity to stimulate the formation of RAS–RAF complexes. Together, this work suggests that compounds that stabilize the NTR-KD interaction and present a high-binding affinity to all RAF paralogs should demonstrate superior RAF inhibitory activity in RAS-mutant cells.

## Results

**RAF inhibitors stimulate RAS–RAF association.** We generated BRET-based biosensors to monitor the physical association between activated KRAS (KRAS[G12V]) and two RAF paralogs (B and CRAF). KRAS constructs comprise an N-terminal *Renilla* luciferase II (RlucII) tag, while full-length RAF constructs were designed as N-terminal GFP[10] fusions[29]. To simplify BRET probe

description, a donor-acceptor (RlucII-GFP$_{10}$) convention will be used throughout the manuscript. In titration experiments where the donor probe level is kept constant, increasing concentrations of the acceptor construct led to a saturation of the KRAS$^{G12V}$-RAF BRET signal, which is indicative of a specific interaction between the probes (Fig. 1a and Supplementary Fig. 1a)[30]. Signal specificity was further demonstrated by its sensitivity to mutations targeting the RBD (BRAF$^{R188L}$ and CRAF$^{R89L}$) or the activation state of RAS (active KRAS$^{G12V}$ vs. inactive KRAS$^{S17N}$) (Fig. 1a and Supplementary Fig. 1a).

We next used this assay to monitor the impact of various RAF inhibitors on the RAS–RAF interaction. For this, we used two types of inhibitors, namely, type I inhibitors that bind the kinase domain in an active-like (DFG-in; helix αC-in) mode and type II inhibitors that bind in an inactive-like (DFG-out and/or helix αC-out) mode[31]. Both type I (GDC-0879 and SB590885) and type II (sorafenib, AZ-628, and PLX4032) inhibitors stimulated KRAS$^{G12V}$–BRAF and KRAS$^{G12V}$–CRAF BRET signals in a dose-dependent manner (Fig. 1b and Supplementary Fig. 1b). In contrast, the KRAS$^{G12V}$–RAF$^{RBD}$ control probe was insensitive to these ATP-competitive inhibitors (Supplementary Fig. 1c, d). Interestingly, distinctions in compound potency and fold-induction could be detected by BRET. For instance, we found that PLX4032 induced a response of lower amplitude compared to the other inhibitors (Fig. 1b and Supplementary Fig. 1b), which is consistent with previous literature[16,20]. Importantly, a MEK inhibitor (U0126) had no impact on KRAS$^{G12V}$–BRAF or KRAS$^{G12V}$–CRAF interactions (Supplementary Fig. 1c, d). This suggested that RAS–RAF complexes induced by RAF inhibitors are not merely caused by inhibiting ERK-mediated negative feedback[32,33].

Given that RAF inhibitors did not impinge on KRAS$^{G12V}$–RAF$^{RBD}$ BRET probes, it strongly suggested that RAF catalytic cleft engagement is required for inhibitor-induced RAS–RAF association. To verify this, we introduced a "gatekeeper" mutation (T529M in BRAF and T421M in CRAF) in RAF BRET probes. This class of mutations prevents type I inhibitor access to the RAF catalytic cleft, but does not affect type II inhibitor binding[20,34]. Consistent with this, the interaction of KRAS$^{G12V}$ with BRAF and CRAF gatekeeper mutants was resistant to GDC-0879 (type I), but not to AZ-628 (type II) in BRET dose-response experiments (Fig. 1c and Supplementary Fig. 1e). These results are in agreement with previous findings whereby the T421M mutation prevented compound-induced CRAF membrane relocalization[16].

To demonstrate by another mean that RAF inhibitors induced RAS–RAF association, we verified their ability to promote the co-immunoprecipitation (co-IP) of endogenous RAF using a Flag-tagged KRAS$^{G12V}$ construct. Six bona fide RAF inhibitors as well as three ATP-competitive inhibitors previously shown to promiscuously bind to RAF were tested[20]. As shown in Fig. 1d, each inhibitor enhanced BRAF and CRAF co-IPs, albeit to various degree. To confirm that this effect was observed regardless of RAS expression levels, we conducted titration experiments and showed that GDC-0879 could induce RAS–RAF association across a wide range of Flag-KRAS$^{G12V}$ amounts (Supplementary Fig. 2a). Also, consistent with the idea that inhibitor-induced RAS–RAF association is not isoform- or allele-specific, we found that NRAS$^{G12V}$, HRAS$^{G12V}$, and KRAS$^{Q61H}$ all exhibited an increased propensity to interact with endogenous BRAF and CRAF upon GDC-0879 treatment (Fig. 1d and Supplementary Fig. 2b). Furthermore, in accordance with the BRET results, MEK inhibitors were inactive in this assay (Fig. 1e). Intriguingly, we noted that PLX4032 and sorafenib only slightly induced RAS–RAF complex formation compared to SB590885, GDC-0879, AZ-628, or dabrafenib (Fig. 1d). Similarly, three non-

selective inhibitors showed a weaker effect on the RAS–RAF interaction (Fig. 1d). This could be explained by the differential affinities or binding modes of the inhibitors. However, although PLX4032 is a poor inducer of the RAS–RAF interaction, it has an in vitro IC$_{50}$ in the same range as GDC-0879 and AZ-628[16,35,36]. Likewise, the binding modes of PLX4032 and dabrafenib are similar (type IIb: DFG-in; helix αC-out), yet they have distinct abilities to drive RAS–RAF association. The same can be said for sorafenib and AZ-628 (type IIa: DFG-out; helix αC-in). Finally, the formation of inhibitor-induced RAS–RAF complexes was recapitulated with endogenous proteins (Supplementary Fig. 2c, d). Together, these findings confirm that RAF inhibitors promote RAS–RAF association and that this event requires RAF catalytic cleft engagement.

**Compound-induced RAS–RAF binding correlates with dimerization.** Given that all RAF inhibitors used above are also known inducers of RAF dimerization, we verified whether their differential ability to impinge on the RAS–RAF association correlated with their dimerization induction potencies. To address this, we treated HEK293T cells with the various compounds and compared their ability to induce endogenous BRAF–CRAF complexes by co-IP. As shown in Fig. 2a, compounds exhibiting strong BRAF–CRAF dimer induction were the same as those demonstrating strong KRAS$^{G12V}$–RAF interaction (Fig. 1d). We quantitatively assessed this correlation by plotting the potency of inhibitors to induce the BRAF–BRAF kinase domain BRET signal against their potency to stimulate the KRAS$^{G12V}$–BRAF interaction (Fig. 2b and Supplementary Fig. 3a, b). The elevated $R^2$ derived from this analysis (0.85) indicates that the inhibitors' potency to stimulate RAS–RAF association indeed strongly correlates with their ability to induce dimerization.

As inhibitor binding influences RAF kinase domain conformation leading to dimerization[20], we reasoned that dimer formation per se might contribute to RAS–RAF complex formation. We sought to address this possibility by using the dimerization-impaired BRAF_R509H mutant allele[7] and tested the ability of distinct type I and type II inhibitors to drive KRAS$^{G12V}$–BRAF association using the BRET assay. Interestingly, while the R509H mutation strongly reduced the potency of type I inhibitors, it only had a mild effect with type II inhibitors (Fig. 2c). Although these findings were consistent with a role for kinase domain dimerization in RAS–RAF complex formation, the divergence in potency observed between the two classes of inhibitors was intriguing. A number of possibilities could explain the difference. For instance, the affinity of the inhibitors for the BRAF kinase domain might be differentially affected by the R509H mutation. Another possibility could be that type II inhibitors are weakly impaired by the R509H mutation in their ability to induce BRAF dimerization. To address the first possibility, we used a time-resolved fluorescence resonance energy transfer (TR-FRET) assay[20] and compared the binding affinity of type I and type II inhibitors to the ATP-binding site of recombinant wild-type (WT) and R509H BRAF kinase domains (Supplementary Fig. 3c and Supplementary Data 1). Unexpectedly, the affinity of type I inhibitors for BRAF$^{R509H}$ was reduced by four to sevenfold compared with WT, whereas type II inhibitors showed equal affinity for WT and BRAF$^{R509H}$ (Fig. 2d). It thus appears that the binding of the two classes of inhibitors is differentially impacted by the R509H mutation. Because of their loss of affinity, type I inhibitors therefore cannot be used in combination with the R509H mutation to unambiguously ascertain the relevance of kinase domain dimerization on RAS–RAF complex formation. We thus turned our attention to type II inhibitors for which the binding affinities were not altered. Although reproducible, the

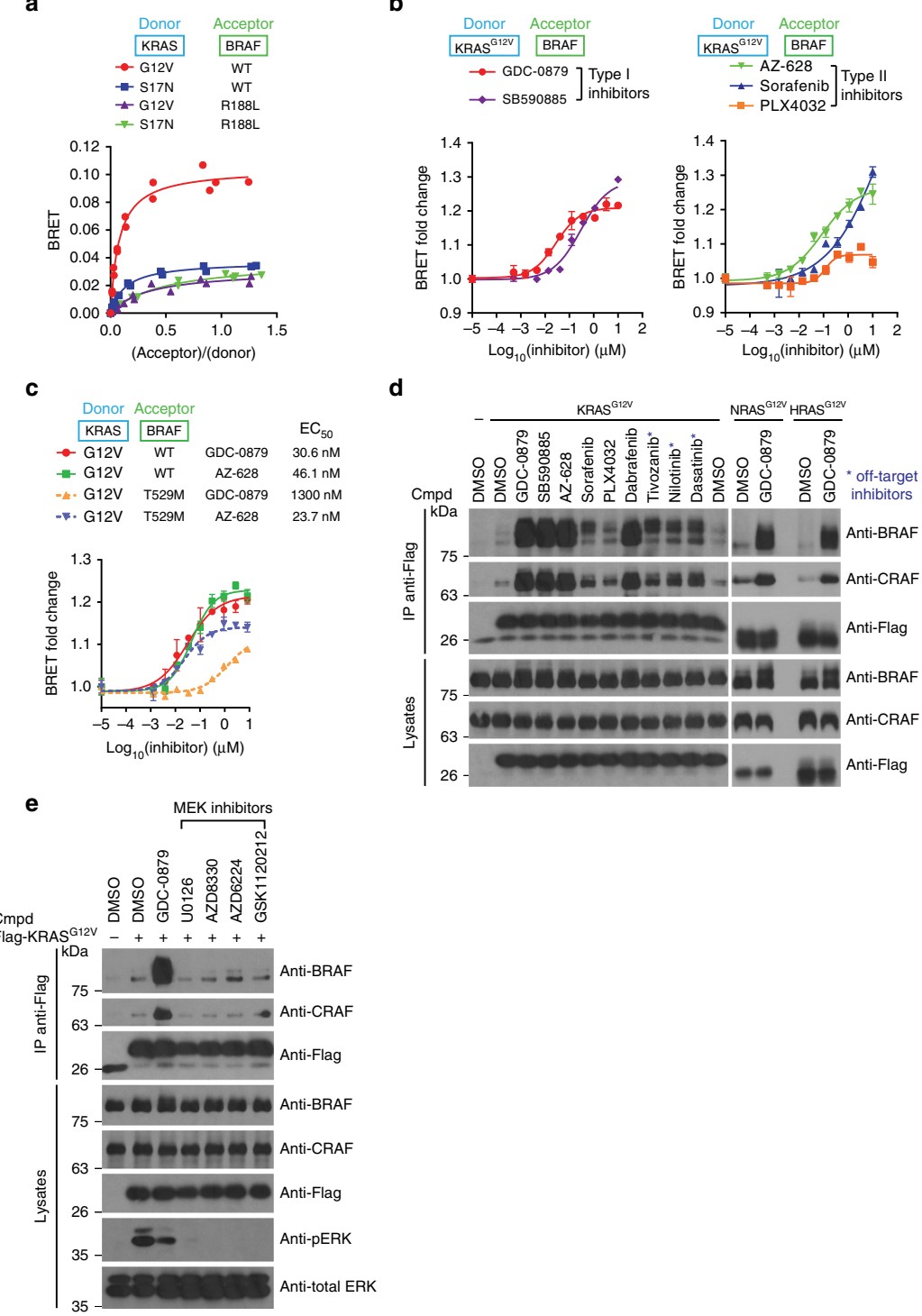

**Fig. 1** RAF inhibitors promote RAS–RAF complex formation. **a** Validation of the KRAS–BRAF BRET biosensor by titration experiments. KRAS donor constructs comprised an N-terminal RlucII tag, while the BRAF acceptor constructs included an N-terminal GFP10 moiety. KRAS$^{G12V}$–BRAF pair produced a strong and reproducible BRET signal that fitted a hyperbolic function, indicating a specific interaction. **b** The BRET signal emitted by the KRAS$^{G12V}$–BRAF biosensors can be induced by RAF inhibitors in a dose-dependent manner. BRET fold-changes are presented separately for type I and type II inhibitors (left and right panel, respectively). **c** Mutation of the BRAF gatekeeper residue (T529M) in BRET probes abolishes GDC-0879 induced, but not AZ-628-induced KRAS$^{G12V}$-BRAF interaction. EC$_{50}$s for each dose-response curve are indicated. **d** Six bona fide RAF inhibitors and three off-target inhibitors enhance the association of Flag-tagged KRAS$^{G12V}$ with endogenous BRAF and CRAF as measured by co-IP (left panel). GDC-0879 treatment also stimulates the association of NRAS$^{G12V}$ and HRAS$^{G12V}$ with endogenous BRAF and CRAF. Cells were treated with 10 μM of the indicated compounds. **e** MEK inhibitors do not induce KRAS$^{G12V}$-BRAF or KRAS$^{G12V}$-CRAF complex formation as measured by co-IP. Cells were treated with 10 μM of the indicated compounds. Error bars in dose-response curves correspond to mean values ± s.d. of technical duplicates of a representative biological triplicate. EC$_{50}$s are the average of at least three independent repeats (Supplementary Data 1)

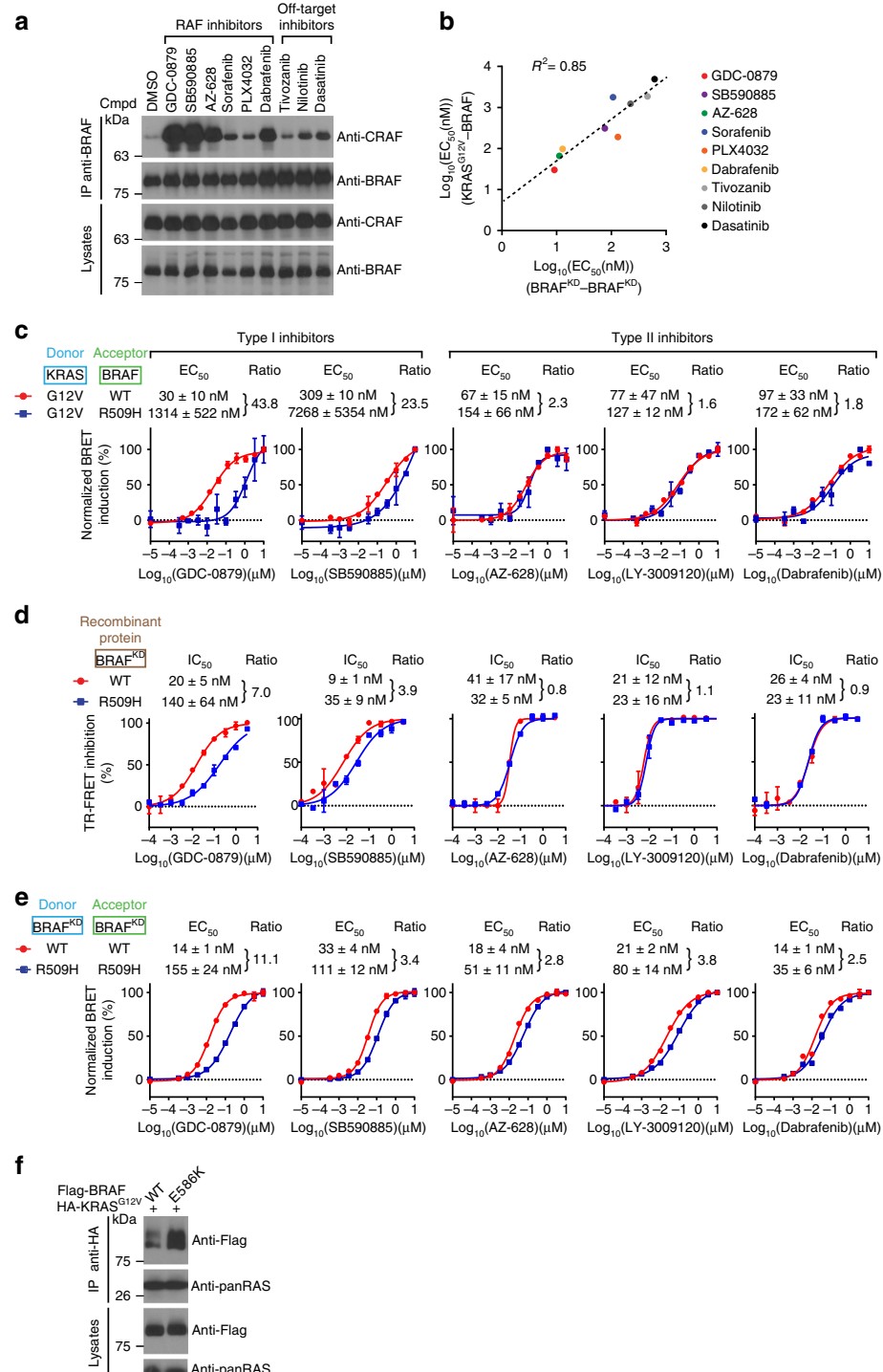

**Fig. 2** Inhibitor-induced RAS−RAF association correlates with RAF dimerization. **a** RAF inhibitors strongly induce the dimerization of endogenous BRAF and CRAF in HEK293T cells as determined by co-IP. Cells were treated with 10 μM of the indicated compounds. **b** Correlation between BRAF kinase domain dimerization and KRAS−BRAF association. $EC_{50}$s obtained for each inhibitor in the BRAF−BRAF kinase domain dimerization assay were plotted against the $EC_{50}$s obtained in the $KRAS^{G12V}$−BRAF BRET assay (Supplementary Fig. 3a, b). **c** The dimerization-impaired $BRAF^{R509H}$ mutant strongly impedes $KRAS^{G12V}$−BRAF association induced by type I inhibitors, but weakly affects induction by type II inhibitors. **d** Binding $IC_{50}$s of representative type I and type II inhibitors determined by TR−FRET using recombinant WT BRAF or $BRAF^{R509H}$ kinase domain. **e** The R509H substitution alters type I and type II inhibitor-induced BRAF kinase domain dimerization to different degrees. **f** The dimerization-enhancing $BRAF^{E586K}$ mutant increases basal $KRAS^{G12V}$−BRAF association. To facilitate comparison between conditions, the range between minimal and maximal BRET signals was normalized to 100% in **c**, **e**. Error bars in dose-response curves correspond to mean values ± s.d. of technical duplicates of a representative biological triplicate. $EC_{50}$s and $IC_{50}$s are the average of at least three independent repeats (Supplementary Data 1)

R509H mutation only weakly impeded the ability of type II inhibitors to drive RAS–RAF association (Fig. 2c). As mentioned above, we addressed whether this could be correlated with a weak effect of the R509H mutation on BRAF dimerization induced by type II inhibitors. As shown in Fig. 2e, we surprisingly found that type II inhibitors were indeed only weakly affected by the R509H mutation. Therefore, although these findings support a model in which kinase domain dimerization plays a positive role in the RAS–RAF association, the lack of appropriate mutants that completely eliminate compound-induced kinase domain dimerization, without compromising the compound's binding affinity, prevents a definitive demonstration.

To address by another means the relevance of kinase domain dimerization in RAS–RAF interaction, we assessed the propensity of a BRAF dimerization-enhancing mutant (E586K)[7] to interact with RAS. BRAF$^{E586K}$ indeed showed an enhanced basal interaction with KRAS$^{G12V}$, which is consistent with a role for dimerization in this event (Fig. 2f).

**RAF inhibitors relieve BRAF autoinhibition**. Our results show that orthosteric binding is a prerequisite for inhibitor-induced RAS–RAF association. This suggests that RAS binding to the RBD is structurally coupled to the kinase domain infrastructure. Notably, an intramolecular interaction has previously been reported between the kinase domain and a portion of the RAF N-terminal regulatory region (NTR) encompassing the RBD and the CRD[4–6,37]. This interaction contributes to RAF autoinhibition in quiescent cells and is relieved upon RAS binding[5]. We reasoned that ATP-competitive RAF inhibitors might perturb this physical interaction and thereby facilitate the access of RAS to the RBD.

To test this hypothesis, we generated BRAF N-terminal regulatory region (residues 1–434; referred to as BRAF$^{NTR}$) and BRAF kinase domain (residues 435–766; referred to as BRAF$^{KD}$) BRET fusions and assessed their interaction in titration experiments. The probes displayed moderate but saturable BRET signals with a close fit to a hyperbolic function (Fig. 3a). Consistent with the ability of GTP-loaded RAS to disrupt RAF intramolecular interaction, the BRET signal was drastically reduced by overexpressing mCherry-tagged KRAS$^{G12V}$, while RBD mutant BRAF$^{NTR}$_R188L was insensitive to RAS expression (Fig. 3a). We next used this assay to examine the impact of inhibitors on RAF intramolecular interactions. Interestingly, dose-response experiments showed that GDC-0879 reduced the NTR–KD BRET signal in a concentration-dependent manner, while MEK inhibitors showed no impact (Fig. 3b).

BRET assays do not only report on complex formation/disruption, but also on conformational changes within protein complexes. We therefore used co-IP as an alternative method to determine whether ATP-competitive inhibitors could also physically disrupt RAF intramolecular interaction. For this, we generated a Polyoma (Pyo) epitope-tagged version of BRAF NTR (Pyo-BRAF$^{NTR}$) and a Flag-tagged version of the kinase domain (Flag-BRAF$^{KD}$). RAF kinases harboring NTR truncations behave as gain-of-functions[38–42] that can be blocked in *trans* by NTR co-expression[4–6]. We recapitulated these findings using Flag-BRAF$^{KD}$ and Pyo-BRAF$^{NTR}$ (Supplementary Fig. 4a). Moreover, BRAF NTR–KD complexes could be detected by co-IP and these were effectively disrupted by co-expressing activated RAS, whereas complexes containing a R188L variant BRAF$^{NTR}$ were resistant (Supplementary Fig. 4b). We then tested the effect of GDC-0879 on these complexes. In agreement with the BRET assay, GDC-0879 reduced BRAF NTR-KD complex formation (Fig. 3c). Interestingly, treatment with GDC-0879 induced a mobility shift of the Flag-BRAF$^{KD}$ protein, which was likely caused by ERK-mediated negative feedback phosphorylation of

BRAF[32,33]. This event was indeed reverted by phosphatase lambda treatment and blocked by an ERK inhibitor (SCH772984)[43] (Supplementary Fig. 4c, d). Importantly, a reduction in BRAF intramolecular interaction upon GDC-0879 treatment was observed even when abrogating ERK-mediated feedback phosphorylation (Supplementary Fig. 4d). We also verified whether disruption of NTR–KD complexes by RAF inhibitors required catalytic cleft binding. For this, we used the gatekeeper mutant BRAF$^{KD}$_T529M in both BRET and co-IP experiments. Consistent with our previous findings (Fig. 1c), this variant showed resistance to GDC-0879-induced disruption of NTR–KD complexes (Supplementary Fig. 4e, f). Finally, to address the role of RAS in compound-induced release of the NTR–KD interaction, we tested the ability of GDC-0879 to disrupt autoinhibition using an NTR construct harboring the R188L substitution and which showed resistance to RAS–GTP (Supplementary Fig. 4b). Strikingly, GDC-0879 equally inhibited the interaction of the kinase domain with WT and R188L mutant BRAF$^{NTR}$ as measured by BRET and co-IP (Fig. 3d, e). Together, these data indicated that the effect of GDC-0879 on RAF autoinhibition depends on compound engagement. In addition, they showed that it is independent of ERK-mediated negative feedback regulation or of RAS binding. This latter observation is particularly significant as it provides a basis to explain the co-operativity observed between RAS binding to the RBD and compound binding to the kinase domain in driving paradoxical activation of the pathway.

**Disruption of autoinhibition correlates with dimerization**. ATP-competitive inhibitor engagement produces a closed/rigid conformation of the RAF kinase domain that stabilizes the side-to-side interface leading to dimerization[20,44]. We reasoned that the compounds' ability to impinge on the NTR-KD interaction might be a consequence of their ability to drive kinase domain dimerization. To investigate this, we calculated the correlation between the potency of nine compounds to induce BRAF dimerization and their potency to impede NTR-KD complexes using BRET EC$_{50}$s as a proxy (Supplementary Figs. 3a, 4g). This analysis generated a $R^2$ of 0.58 (Fig. 3f). Although consistent with a connection between the two events, their mild correlation suggests that compound-induced kinase domain dimerization might not be the sole factor impacting the NTR-KD interaction. To further investigate the relevance of compound-induced dimerization in NTR-KD disruption, we tested by BRET the effect of the R509H mutation on the ability of type I and type II inhibitors to alter the NTR-KD interaction with the caveat that type I inhibitors bind with lower affinity the BRAF_R509H kinase domain (Fig. 2d). Type I inhibitors were indeed less effective at disrupting the NTR-KD_R509H interaction (Fig. 3g), which might result from their reduced affinity for BRAF_R509H. In marked contrast, type II inhibitors were not affected by the R509H mutation. Given their normal binding affinity to BRAF_R509H, it suggests that dimerization per se plays a minor role if any in the ability of RAF inhibitors to disrupt the NTR-KD interaction. This conclusion, however, has to be taken cautiously as type II inhibitors can still induce, albeit with a two to fourfold reduced potency, the dimerization of BRAF_R509H (Fig. 2e).

Based on the above findings, we reasoned that compound occupancy of the kinase domain cleft and its specific effect on the kinase domain conformation could alter in *cis* the NTR–KD interaction, and in turn, physically abrogates the interaction or merely modifies it without necessarily disrupting it. As a result, such events would influence the accessibility of the RAF RBD for RAS–GTP. Compounds like AZ-628, which is a potent inducer of dimerization but a poorer disruptor of the NTR–KD interaction,

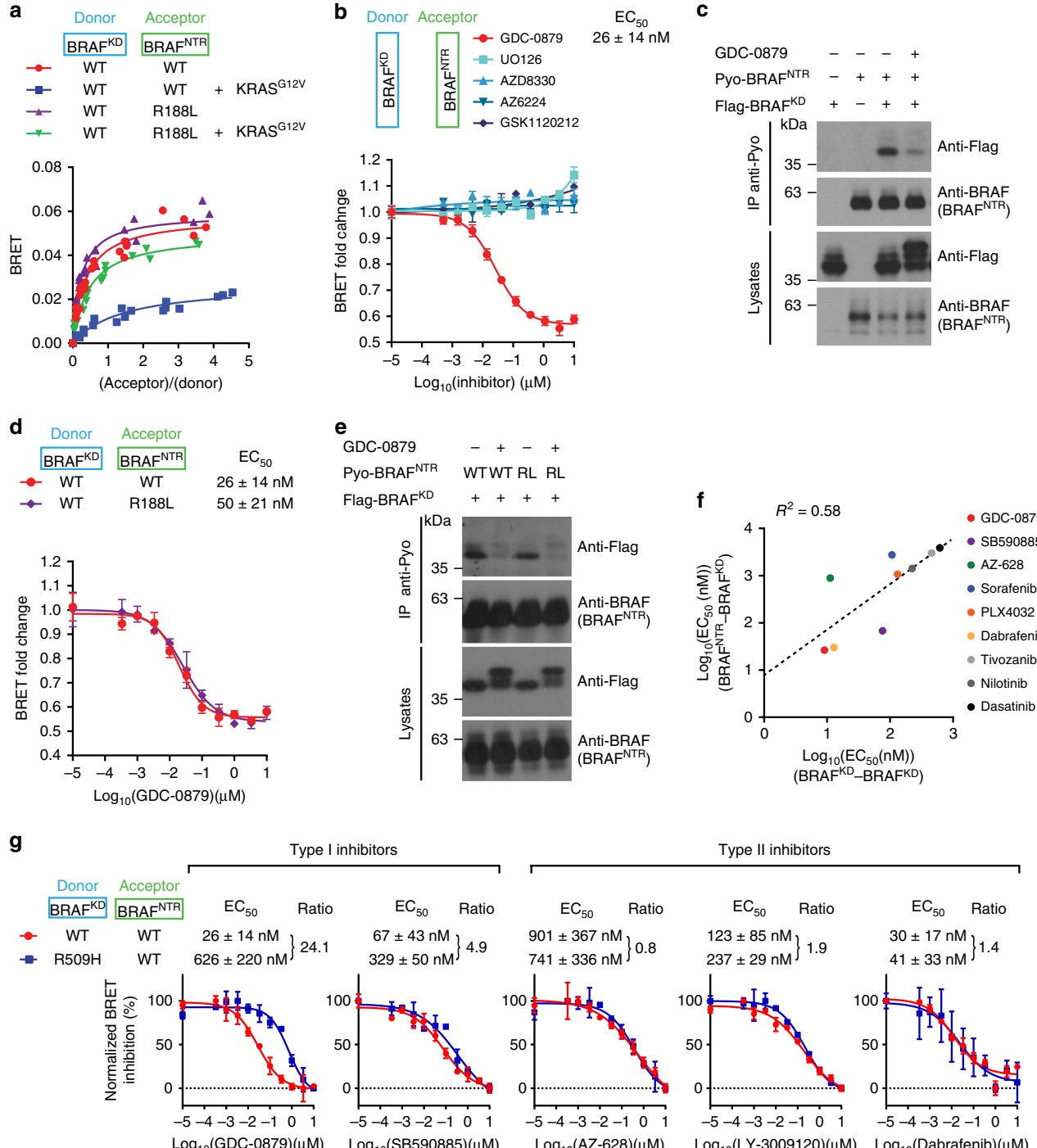

**Fig. 3** RAF ON-state inhibitors disrupt BRAF intramolecular interaction in a RAS-independent manner. **a** BRET titration curves demonstrating the specificity of BRAF intramolecular biosensors (BRAF$^{NTR}$–BRAF$^{KD}$). Addition of mCherry-KRAS$^{G12V}$ robustly impaired the BRAF NTR-KD association, whereas a probe containing the BRAF$^{NTR}$_R188L mutant was insensitive to activated RAS. **b** Dose-response analysis of the BRAF$^{NTR}$–BRAF$^{KD}$ BRET probes with the RAF inhibitor GDC-0879 and a series of MEK inhibitors. **c** GDC-0879 disrupted the BRAF$^{NTR}$–BRAF$^{KD}$ complex in co-IP experiments. Anti-BRAF was used to detect the expression of Pyo-tagged BRAF$^{NTR}$. BRET **d** and co-IP **e** show that GDC-0879 equally disrupts the interaction of the BRAF kinase domain with either WT or R188L (RL)-mutated NTR constructs. **f** Correlation between inhibitor-induced BRAF kinase domain dimerization and the disruption of BRAF NTR-KD interaction. EC$_{50}$s for each inhibitor in the BRAF-BRAF kinase domain dimerization assay were plotted against EC$_{50}$s obtained with the BRAF$^{NTR}$-BRAF$^{KD}$ BRET assay. **g** The dimerization-impaired mutant BRAF$^{R509H}$ impedes BRAF$^{NTR}$–BRAF$^{KD}$ disruption induced by type I inhibitors, but does not significantly impede disruption induced by type II inhibitors. GDC-0879 was used at 1 µM in all co-IP experiments. EC$_{50}$s are the average of at least three independent replicates (Supplementary Figs. 3a, 4g; Supplementary Data 1). To facilitate comparison between conditions, the range between minimal and maximal BRET signals was normalized to 100% in **g**. Error bars in dose-response curves correspond to mean values ± s.d. of technical duplicates of a representative biological triplicate. EC$_{50}$s are the average of at least three independent repeats (Supplementary Data 1)

could be of the latter category (Fig. 3f; Supplementary Fig. 4g). In support for this model, we found that several BRAF oncogenic mutations exhibit a clearly enhanced interaction with KRAS$^{G12V}$ (Supplementary Fig. 5a), yet their respective NTR–KD interaction is comparable to that of WT BRAF (Supplementary Fig. 5b) as if physical NTR disengagement from the kinase domain is not essential for increased access of RAS–GTP to the RBD. Be that as it may, the NTR–KD interaction of BRAF oncogenic variants such as V600E or G469V was nonetheless physically impaired upon compound treatment (Supplementary Fig. 5c), which indicates that inhibitors alter the NTR–KD interaction in BRAF-mutant cells.

**RAF inhibitors selectively alter RAF autoinhibition.** Given the related structural organization of RAF family kinases, we wondered whether the release of the NTR–KD interaction observed in BRAF could also occur in other family members. To address this, we first determined the ability of the NTR of each RAF and KSR isoform to interact with its cognate kinase domain. In addition to BRAF and CRAF, which have been previously reported[4,5], we detected NTR–KD association also for ARAF, KSR1, and KSR2 (Fig. 4a). These findings support the notion that intramolecular binding is a common regulatory feature of the RAF family members. Interestingly, we noticed considerable differences in interaction strength between the different isoforms, but this did not correlate with their apparent intrinsic kinase activity (Supplementary Fig. 6).

Next, we examined how each NTR–KD interaction responded to a type I (GDC-0879) or a type II (LY-3009120) inhibitor. In addition to BRAF, the CRAF NTR–KD interaction was also disrupted in response to drug treatment, whereas ARAF, KSR1, and KSR2 showed no response (Fig. 4b). The absence of response of the ARAF NTR–KD interaction to the inhibitors was surprising, especially for LY-3009120 since it has been reported to bind ARAF in vitro with an affinity similar to BRAF or CRAF[22]. This may imply that ARAF conformational response to compounds is distinct from that of BRAF and CRAF.

**RAF inhibitors and RAS co-operate to induce RAF dimers.** Under physiological conditions, RAS-GTP binding is the primary trigger implementing the relief of autoinhibition, which subsequently leads to RAF dimerization. Previous work has shown that RAF inhibitors require RAS activity in order to induce ERK signaling[16–18]. Our analysis identified BRAF and CRAF as the only isoforms sensitive to compound-induced NTR–KD dissociation. In addition, we found that this phenomenon takes place independently of RAS activity. Since NTR release by RAS stabilizes the ON-state of the kinase domain through dimerization, we hypothesized that RAF inhibitors might similarly induce the dimerization of full-length RAF proteins independently of RAS activity, owing to the action of two concurring events, namely, the ability of inhibitors to alleviate the NTR–KD interaction and their ability to stabilize a closed/active-like conformation of the kinase domain. This phenomenon would nevertheless occur at a low rate given the absence of RAS–GTP that effectively relieves autoinhibition and nucleates the formation of RAF dimers through nanoclustering at the plasma membrane[5,28].

To test this hypothesis, we assessed the ability of GDC-0879 and LY-3009120 to induce BRAF–CRAF dimerization ± RAS activity. For this, we conducted BRET dose-response experiments using WT or RBD-inactivated (RL variants) BRAF and CRAF probes expressed alone or together with KRAS$^{S17N}$ or KRAS$^{G12V}$ (Supplementary Fig. 7a). Consistent with our prediction, compound-induced BRAF–CRAF dimerization was observed even in conditions where no RAS activity was available or when

RAF proteins were prevented from associating with RAS–GTP (Fig. 4c). We made similar observations with additional RAF inhibitors (Supplementary Fig. 7b). However, the compounds systematically showed much lower potencies under these conditions than when WT RAF probes were co-expressed along with KRAS$^{G12V}$ (Fig. 4c and Supplementary Fig. 7b). Similar findings were obtained by co-IP using co-expressed RAF$^{RL}$ constructs (Fig. 4d). It thus appears that RAF inhibitors can induce RAF dimerization independently of RAS activity.

Given the independent action of RAF inhibitors and RAS activity in promoting RAF conformational transitions, we surmised that both inputs might work co-operatively, thus explaining their potent combinatorial effects. To address this, we took advantage of the BRAF–CRAF BRET interaction assay and conducted dose-response experiments in cells expressing varying amounts of the mCherry-RAS$^{G12V}$ construct. Data analysis allowed to calculate logα's (measure of cooperativity[45]) of 1.76 and 1.28 for GDC-0879 and LY-3009120, respectively. This indicated that compounds and RAS indeed work co-operatively and it confirmed that RAS–GTP is a positive allosteric modulator of compound-induced RAF dimerization.

**Effect of second-generation inhibitors on RAF interactions.** Given the shortcomings of initial RAF inhibitors, a new generation of compounds has recently emerged comprising two main classes defined by their mechanism of action (Introduction), namely, pan-RAF inhibitors and "paradox breakers" (also referred to as PBs)[26]. We sought to compare these two classes of inhibitors with respect to their impact on RAF autoinhibition and RAS–RAF association. For this, we used the pan-RAF inhibitor LY-3009120 and a representative paradox breaker, PLX-0012 (WO 2012109075, compound P-0012)[46], which belongs to the same chemotype as PLX4720, but features a N,N-methyl-ethyl-sulfamoyl moiety instead of the n-propyl-sulfonamide group. This subtle difference between the two compounds was claimed to confer paradox-breaking properties to sulfonamide series inhibitors (Fig. 5a)[26].

We first profiled these inhibitors in our panel of BRET probes. LY-3009120 strongly and potently induced RAF dimerization in the nM range (Fig. 5b; Supplementary Fig. 8a, b). It also strongly stimulated KRAS–BRAF and KRAS–CRAF interactions and inhibited RAF intramolecular interaction with potencies similar to GDC-0879 (Fig. 5d, f; Supplementary Fig. 8c). Under these criteria, LY-3009120 is thus comparable to first-generation RAF inhibitors. Co-IP experiments confirmed that LY-3009120 strongly induces RAF dimerization, promotes the formation of RAS–RAF complexes and disrupts RAF autoinhibition (Fig. 5c, e, and g). In stark contrast, PLX-0012 did not influence BRAF–BRAF dimerization (Fig. 5b) while it only slightly induced CRAF–CRAF and BRAF–CRAF dimerization at elevated doses (Supplementary Fig. 8a, b). Similarly, PLX-0012 only induced a mild increase in KRAS–RAF association (Fig. 5d; Supplementary Fig. 8c), and had a minor effect on the NTR–KD interaction (Fig. 5f). Notably, the effect of PLX-0012 on each probe was weaker than PLX4720. Co-IP experiments confirmed these BRET results (Fig. 5c, e and g; Supplementary Fig. 8d).

Next, we assessed whether the profile of these inhibitors in BRET and co-IP assays predicted their effect on ERK signaling in a panel of cell lines harboring RAS mutations (Table 1; Supplementary Fig. 9, Supplementary Data 1). The BRAF$^{V600E}$ mutant melanoma cell line A375 was included as a reference in which, all three compounds fully and potently suppressed pERK (Table 1; Supplementary Fig. 9g). Consistent with a BRET profile similar to second-generation compounds, LY-3009120 induced pERK activation with variable potencies in each KRAS-mutant

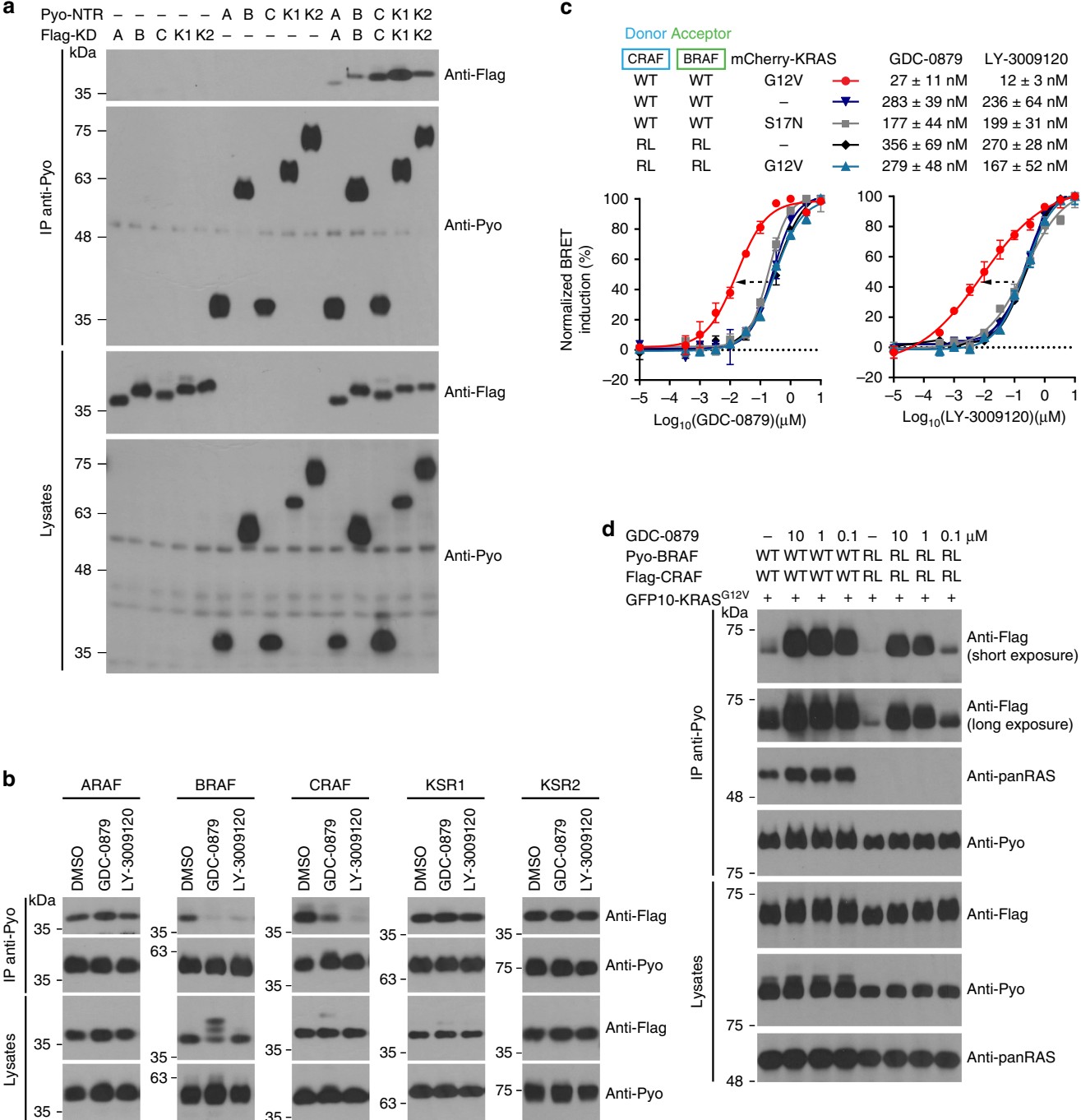

**Fig. 4** ON-state RAF inhibitors selectively disrupt BRAF and CRAF intramolecular interactions. **a** The NTR of each RAF family member interacts specifically with its cognate kinase domain. The strength of the interaction monitored by co-IP varied across isoforms. The NTR of each family member (ARAF (A), BRAF (B), CRAF (C), KSR1 (K1), and KSR2 (K2)) was fused with a N-terminal Pyo tag, while its cognate kinase domain (KD) harbored a N-terminal Flag tag. **b** The RAF inhibitors GDC-0879 and LY-3009120 selectively disrupt the NTR–KD interaction of BRAF and CRAF, while they had no detectable effect on the other family members (ARAF, KSR1, and KSR2). Cells were treated with 10 μM of GDC-0879 or LY-3009120. **c** RAF inhibitors induce BRAF–CRAF dimerization in a RAS-independent manner. GDC-0879 (left) and LY-3009120 (right) induce full-length CRAF-BRAF dimerization in the presence of dominant-negative $KRAS^{S17N}$ (gray) or RBD-mutated $BRAF^{R188L}$ and $CRAF^{R89L}$ (RL, blue curves). Expression of $KRAS^{G12V}$ potentiates (dashed arrows) the effect of both compounds on BRAF–CRAF dimerization (red curves). **d** Co-IP confirms that $BRAF^{R188L}$ and $CRAF^{R89L}$ can form inhibitor-induced, RAS-independent dimers. To facilitate comparison between conditions, the range between minimal and maximal BRET signals was normalized to 100% in **c**. Error bars in dose-response curves correspond to mean values ± s.d. of technical duplicates of a representative biological triplicate. $EC_{50}$s are the average of at least three independent repeats (Supplementary Data 1)

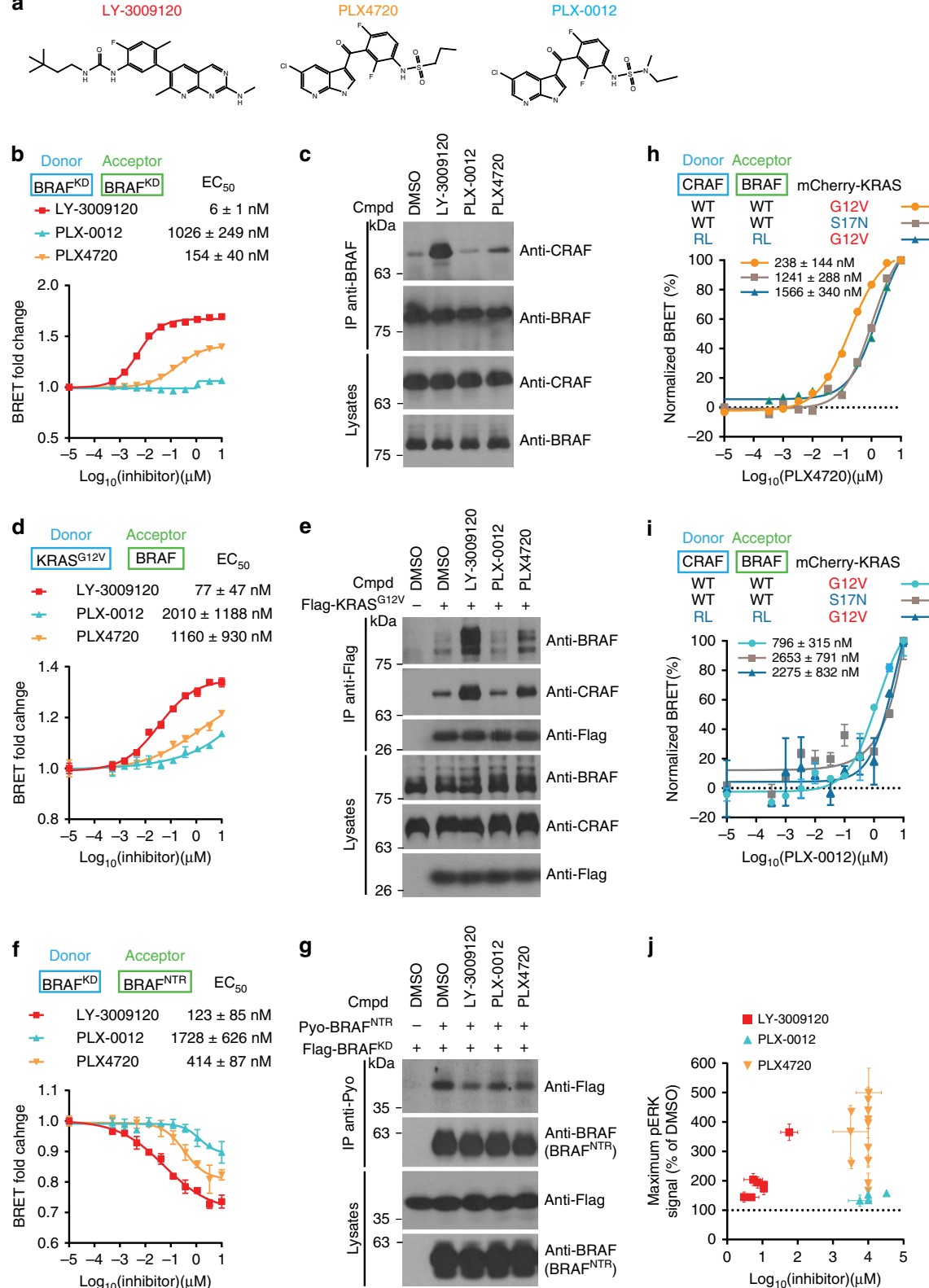

cancer cell line examined (Table 1). As reported previously, PLX4720 led to pERK induction in all tested KRAS-mutant lines (Table 1). In contrast, PLX-0012 did not produce strong paradoxical pERK induction across eight KRAS$^{mut}$ lines (Table 1 and Supplementary Fig. 9). However, we noticed that, despite this improved profile, PLX-0012 caused a slight but consistent pERK induction in SW900 ($125 \pm 23\%$ for AlphaLISA SureFire Ultra and $158 \pm 8\%$ for MSD technology). This could be linked to the fact that despite much weaker effects, PLX-0012 nonetheless exhibited a propensity at higher doses to stimulate RAF dimers and RAS–RAF association, as well as to block NTR–KD interaction (Fig. 5 and Supplementary Fig. 9). Accordingly, compared to PLX4720, PLX-0012 showed a reduced potency to stimulate RAS-independent CRAF–BRAF dimerization and, in agreement with its improved profile in RAS-mutant cells, KRAS$^{G12V}$ only weakly potentiated PLX-0012 effect on full-length BRAF–CRAF dimerization (compare Fig. 5h, i). Our profiling data therefore suggests that LY-3009120 can still stimulate modest paradoxical pathway activation at relatively low doses despite its improved pan-RAF binding potency and thus could represent a liability against RAS-dependent tumors (Fig. 5j and Table 1). It also confirms that PBs represent a substantial improvement over first-generation sulfonamides against BRAF$^{V600E}$ tumors. Nevertheless, PLX-0012 treatment still led to weak paradoxical pathway activation in a limited set of cell lines (Fig. 5j and Table 1). The higher dose required to achieve this effect; however, might be less of a concern. Finally, their poor efficacy in RAS-mutant cells currently confines PB molecules to treat BRAF$^{V600E}$ tumors.

## Discussion

Here, we investigated the impact of small molecule inhibitor binding on the RAF holoenzyme. We found that ATP-competitive binders that stabilize the ON-state conformation of the kinase domain promote RAS–RAF association by disrupting RAF kinase domain autoinhibition mediated by the N-terminal regulatory region (NTR) (Fig. 6a). This suggests that paradoxical ERK activation mediated by RAF inhibitors does not solely result from induced kinase domain dimerization, but also from their ability to alleviate the RAF intramolecular autoinhibited state.

Compound-induced RAS–RAF association has previously been observed but the structural underpinning is not understood[16,27]. Karoulia et al.[21] recently reported that RAF dimerization induced by inhibitors requires RAS–GTP binding to RAF. Interestingly, they correlated the propensity of inhibitors to induce dimerization and RAS–RAF binding to the compounds' ability to re-orient the side chain of the R506 residue situated nearby the dimerization interface at the C-terminal end of helix αC. They suggested that this area might play a role in the NTR–KD interaction; however, this remains to be experimentally investigated. Our data goes one step further by showing that ON-state RAF inhibitors physically disrupt the NTR–KD interaction and that this occurs independently of RAS activity. We propose that this phenomenon explains the positive co-operativity observed between RAS–GTP and RAF inhibitors in driving dimerization and paradoxical RAF activation.

Consistent with published work, we demonstrated a strong correlation between the ability of RAF inhibitors to promote RAS–RAF interaction and kinase domain dimerization (Fig. 2b)[21]. Furthermore, our results suggest that compound-induced dimerization plays a role in RAS–RAF interaction. However, they do not allow to conclude whether dimerization is essential or merely one component of a more complex process underlying RAF holoenzyme assembly. The weaker correlation observed between compound-induced NTR–KD disruption and kinase domain dimerization (Fig. 3f) supports the notion that dimerization is indeed not the sole factor explaining the diversity of RAF inhibitors' effect on this complex. We suggest that conformational changes that occur in cis upon inhibitor binding to the RAF catalytic cleft are also involved in the disruption of the NTR–KD interaction. These structural changes may very well be connected to the alignment of kinase domain hydrophobic spines induced by compound binding[20]. These in cis conformational changes might relieve RAF autoinhibition and lead to RAS-independent dimerization of the kinase domain. This in turn would strengthen the association of RAF with RAS–GTP nanoclusters at the plasma membrane. Finally, amplification of nanocluster formation by RAF dimerization[28] would implement a feedforward loop contributing to the cooperative behavior of the system (Figs. 4c, 6a). Despite that this model accounts for several empirical observations, further work and new technologies will be required to unambiguously ascertain the order of RAS-mediated RAF holoenzyme assembly upon RAF inhibitor treatment.

There is presently no structural information describing the NTR–KD interaction. The finding that each RAF family member undergoes this interaction, albeit with distinct affinities (Fig. 4a), suggests a conserved mechanism. Incidentally, this may also indicate that ARAF, KSR1, and KSR2 are subjected to auto-inhibition, but this will necessitate further investigation. Another aspect requiring structural clarification is the impact of the NTR–KD interaction on RAS–GTP binding to the NTR. Increased RAS–RAF association upon drug-binding suggests that the kinase domain allosterically controls the access of RAS to the NTR. Two simple scenarios can be envisioned: (1) the kinase domain might partially occlude the RBD by interacting with specific residues involved in the RAS–RAF interaction; (2) the NTR–KD interaction might allosterically modulate the affinity of the RBD for RAS.

First-generation ON-state RAF inhibitors potently stabilize kinase domain closure, stimulate formation of dimers, and

**Fig. 5** Profiling of the pan-RAF inhibitor LY-3009120 and the paradox breaker PLX-0012. **a** Structure of LY-3009120, PLX4720, and the Paradox Breaker PLX-0012. Dose-response analysis of LY-3009120, PLX4720 and PLX-0012 with BRET biosensors measuring BRAF kinase domain dimerization **b**, KRAS-BRAF association **d**, and BRAF intramolecular interaction **f**. LY-3009120, PLX4720, and PLX-0012 have distinct propensities to promote BRAF–CRAF dimerization **c**, to stimulate KRAS-RAF association **e**, and to perturb BRAF autoinhibition **g** as measured by co-IP experiments. Cells were treated with 10 μM of each indicated compound. PLX4720 **h** and PLX-0012 **i** induce CRAF–BRAF dimerization in a RAS-independent manner. Full-length BRAF–CRAF dimerization was measured in the presence of dominant-negative KRAS$^{S17N}$ (gray) or RBD-mutated BRAF$^{R188L}$ and CRAF$^{R89L}$ (RL; blue). Expression of active KRAS$^{G12V}$ potentiated the effect of PLX4720 and PLX-0012 on BRAF–CRAF dimerization (orange and cyan, respectively). Each EC$_{50}$ was the average of three independent replicates. **j** Maximum pERK signal reported in Table 1 was plotted against its corresponding concentration in each cell line. Each compound is represented by a distinct color: LY-3009120 (red), PLX4720 (orange), and PLX-0012 (blue). To facilitate comparison between conditions, the range between minimal and maximal BRET signals was normalized to 100% in panels h and i. Error bars in dose-response curves correspond to mean values ± s.d. of technical duplicates of a representative biological triplicate. EC$_{50}$s are the average of at least three independent repeats (Supplementary Data 1). Error bars in **j** correspond to mean values ± s.d. of biological triplicates

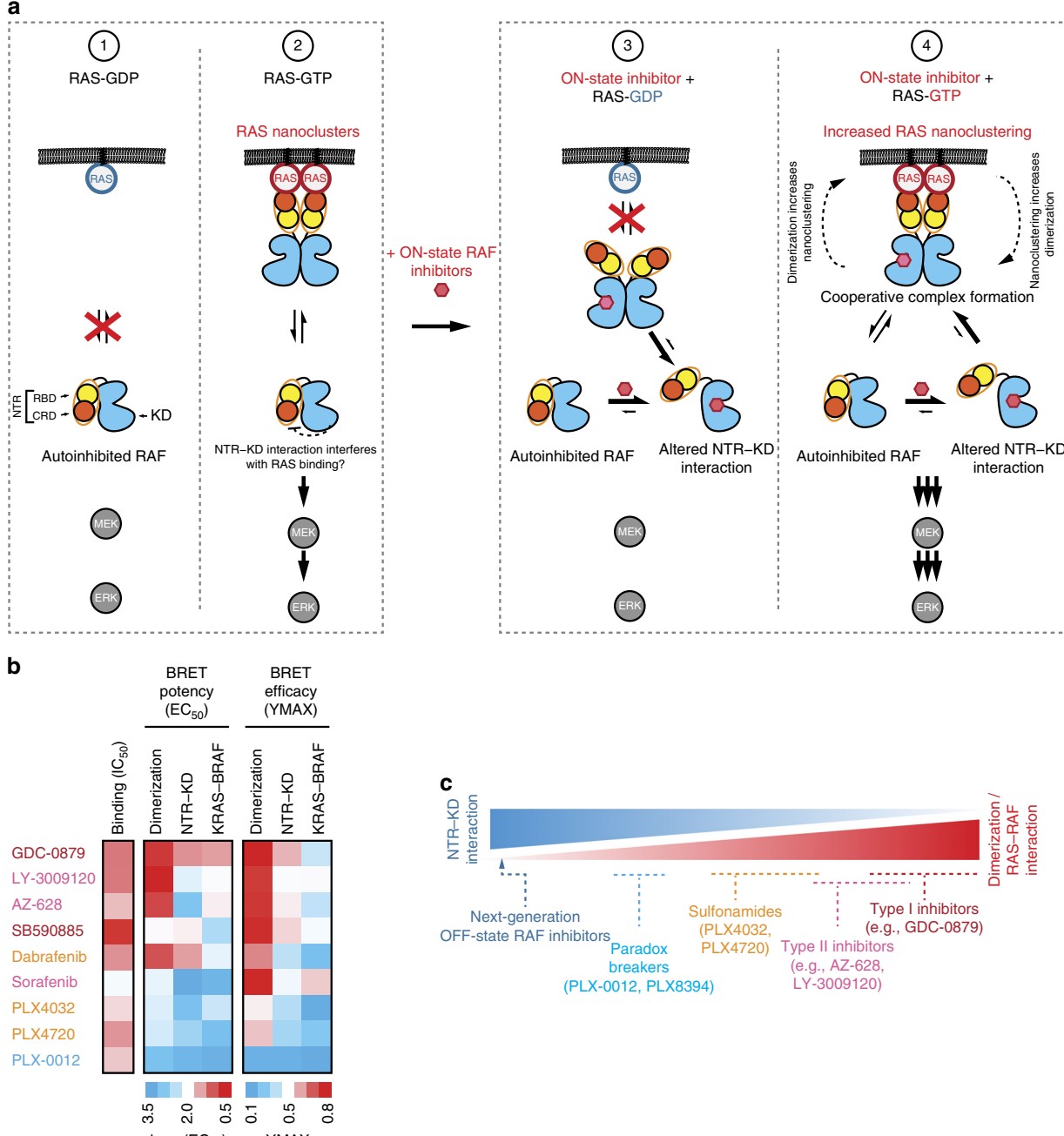

**Fig. 6** Mode of action of activating RAF inhibitors and proposed characteristics of next-generation RAF inhibitors. **a** Model depicting the biological mode of action of RAF inhibitors in untreated (1 and 2; left) or inhibitor-treated cells (3 and 4; right). (1) In quiescent cells where RAS activity is low, RAF proteins are autoinhibited through an interaction between their N-terminal regulatory region (NTR) and their kinase domain (KD). Upon physiological RAS activation or in RAS-activated cancer cells (2), RAF proteins shuttle between an autoinhibited monomeric state in the cytoplasm and an activated dimeric state bound to RAS−GTP at the plasma membrane. At steady-state, ERK signaling is moderated by this ON/OFF equilibrium. (3) ON-state RAF inhibitors disrupt the normal NTR−KD interaction. In cells with low-RAS activity, derepressed RAF can dimerize but this process is not stabilized by active RAS and therefore does not lead to paradoxical pathway activation. (4) RAS activity and RAF inhibitors co-operatively promote RAS−RAF complex formation and RAF dimerization. On the one hand, NTR−KD disruption is triggered by two separate means (RAS-binding to the RBD and compound binding to the catalytic cleft) resulting in dissociation. On the other hand, the increase in RAF effective concentration when bound to RAS nanoclusters coupled with the closed/active-like kinase domain conformation stabilized by the inhibitors leads to increased side-to-side RAF dimerization. In turn, RAF dimerization further augments the population of RAS nanoclusters. Together, these events co-operatively drive RAS−RAF complex formation and RAF dimerization. At sub-saturating compound concentrations, this phenomenon leads to catalytic transactivation of the compound-free protomer. **b** Unbiased clustering of RAF inhibitors based on their effects on dimerization, NTR−KD interaction and RAS−RAF association (BRET EC$_{50}$s and YMAXs values served as clustering parameters). The EC$_{50}$s and YMAXs used for clustering were the average of three independent repeats (Supplementary Data 1). **c** Each class of RAF inhibitors is positioned along an axis from ON-state (red) to OFF-state compounds (blue). Ideal (OFF-state) RAF inhibitors would not allow kinase domain dimerization, but would stabilize the NTR−KD interaction, thereby maintaining RAF proteins in an autoinhibited state irrespective of the status of RAS activity

**Table 1 Phospho-ERK profiling of LY-3009120, PLX4720, and PLX-0012 paradox breaker in a panel of KRAS-mutant cancer cell lines**

| | Cell line | Mutation | LY-3009120 | | PLX4720 | | PLX-0012 | |
|---|---|---|---|---|---|---|---|---|
| | | | Max. pERK signal (%)[b] | Conc. (μM)[c] | Max. pERK signal (%)[b] | Conc. (μM)[c] | Max. pERK signal (%)[b] | Conc. (μM)[c] |
| KRAS-mutant | PANC10.05[a] | p.G12D | 293 ± 35 | 0.033 | 764 ± 316 | 10 | No induction | |
| | SW480[a] | p.G12V | 190 ± 13 | 0.003 | 494 ± 16 | 3.3 | No induction | |
| | SW900[a] | p.G12V | 186 ± 25 | 0.003 | 300 ± 56 | 10 | 125 ± 23 | 3.3 |
| | Calu-6[a] | p.Q61K | 195 ± 34 | 0.003 | 264 ± 25 | 3.3 | No induction[d] | |
| | MIA-PaCa-2 | p.G12C | 212 ± 7 | 0.010 | 491 ± 21 | 10 | No induction | |
| | A427 | p.G12D | 130 ± 7 | 0.001 | 271 ± 53 | 3.3 | No induction | |
| | LOVO | p.G13D | 147 ± 6 | 0.003 | 273 ± 46 | 3.3 | No induction | |
| | HCT 116 | p.G13D | 168 ± 4 | 0.003 | 374 ± 38 | 3.3 | No induction | |
| BRAF-mutant | A375[e] | p.V600E | $IC_{50}$: 0.005 | | $IC_{50}$: 0.011 | | $IC_{50}$: 0.050 | |

[a]The pERK signals were also assessed by Meso Scale Discovery technology (Supplementary Fig. 9d, f)
[b]Cutoff for pERK induction was set at 125% pERK signal
[c]The indicated concentration corresponds to the amount of compound that induced maximal pERK signal
[d]No induction: maximal pERK signal (%) was below 125%
[e]$IC_{50}$s for pERK inhibition are indicated for the BRAF[V600E] mutant A375 cell line
Raw data are reported in Supplementary Fig. 9

enhance RAS–RAF association[20]. We found that those events are also linked to the disruption of RAF autoinhibited complexes. However, one class of compounds, the n-propyl-sulfonamide series, deviated from this profile. These inhibitors (e.g., vemurafenib and PLX4720) have a reduced propensity to disrupt RAF autoinhibition (this work) and consistently have a marginal impact on RAF membrane recruitment[16]. Moreover, these compounds poorly induce BRAF–CRAF heterodimerization in cells and inhibit BRAF kinase domain homodimerization as measured in vitro using purified proteins[16,47]. This property allowed us to obtain the first monomeric crystal structure of the BRAF kinase domain[47]. Yet, other crystallographic data with n-propyl-sulfonamides and phenyl-sulfonamides (e.g., dabrafenib), which share a helix αC-out binding mode, demonstrated that they retained the capacity to promote a dimeric state of the BRAF kinase domain[26,48]. Therefore, it appears that depending on the conditions, sulfonamides have the ability to stabilize either monomers or dimers, which in turn still leads to paradoxical pathway activation in RAS-mutant cancer cells.

Plexxikon recently exploited this unique binding mode of sulfonamides to develop PBs[26]. These compounds were screened for their efficacy at blocking phospho-ERK in BRAF[V600E] tumor cells, while maintaining an induction-free profile in RAS-mutant cells[26]. Like n-propyl-sulfonamides, the N-ethyl-N-methylsulfamoyl amide moiety of these compounds interacts with helix αC leucine 505. Comparison of these two series showed that, instead of drastically impacting the conformation of the RAF kinase domain, N-ethyl-N-methyl-sulfamoyl amides merely induced a subtle change in the conformation of helix αC. This was proposed to further disrupt the RAF ON-state thereby explaining PBs biological activity[26]. While our evaluation of a representative paradox breaker (PLX-0012) did indeed reveal an improved mechanistic profile over n-propyl-sulfonamides (Fig. 5), we still noted weak but detectable induction of RAF dimers, RAS–RAF association, and a marginal reduction of BRAF NTR–KD interaction with PLX-0012. These observations are consistent with the fact that the RAF kinase domain maintains a dimeric state under crystallographic conditions when bound to this series of inhibitors[26]. In agreement with PLX-0012 profile, we observed a weak paradoxical induction in one out of eight KRAS-mutant cancer cell lines. PBs show great improvements over n-propyl-sulfonamides in terms of mechanistic and biological profile (Fig. 6b, c),

which is ideal for targeting BRAF[V600E] mutant cells. Yet, they are poor ERK signaling inhibitors in RAS-mutant cells. One likely explanation is that they do not effectively bind the second protomer of RAF dimers due to negative co-operativity[11,21].

Paradoxical pathway induction was shown to occur through the induction of A-, B- and, CRAF homo- and heterodimers at sub-saturating doses of inhibitors[16–18]. An approach to blunt paradoxical pathway activation was therefore to saturate the cellular pool of RAF with high-affinity inhibitors. These mostly comprise type II molecules that bind RAF isoforms in the low nM range (e.g., AZ-628, LY-3009120, TAK-632, BGB-283)[22,23,25,49]. Interestingly, X-ray co-crystal structures revealed that these compounds stabilize a typical ON-state conformation of the kinase domain[22,25]. This is made possible by the ability of type II compounds to chemically substitute, via a hydrophobic moiety, the displaced R-spine residue Phe595[20]. As expected from this binding mode, pan-RAF inhibitors displayed a mechanistic profile similar to that of first-generation ON-state RAF inhibitors: induction of dimerization; disruption of NTR–KD interaction; and stimulation of RAS–RAF association (Fig. 6b, c). Consequently, paradoxical pathway activation by pan-RAF inhibitors, exemplified by LY-3009120, is still detectable in KRAS-mutant cancer cells (Supplementary Fig. 9 and ref. [22]). This suggests that enhanced binding affinity does not fully eliminate the therapeutic concerns associated with these new inhibitors. Whether this will lead to adverse effects in patients remains to be seen.

In summary, we propose that inhibitor-induced RAF activation is mediated by three interconnected events, namely, NTR–KD complex disruption, RAS–RAF association, and kinase domain dimerization. Conformational coupling and co-operativity between these three parameters is probably one of the main challenges for the development of effective RAF inhibitors targeting RAS-mutant tumors. Our work provides a conceptual framework and screening tools for such molecules. Indeed, it is instructive to consider the behavior of current inhibitors with respect to the three aforementioned events (Fig. 6b, c) as it suggests that small molecules that would strengthen the NTR–KD intramolecular interaction, while at the same time bind with high affinity and stabilize the kinase domain in its monomeric OFF-state, is a path forward for the next generation of RAF inhibitors.

## Methods

**Plasmids and reagents**. All constructs were inserted in pCDNA3.1-Hygro backbone (Invitrogen). BRET fusions were generated by inserting coding sequences of full-length human BRAF, CRAF, and KRAS or relevant truncations (BRAF[1–434], BRAF[RBD] (amino acids 146–237), BRAF[435–766], CRAF[RBD] (amino acids 51–131)) between KpnI and XbaI in plasmids already containing a N-terminal or C-terminal cassette encoding GFP$_{10}$ or *Renilla* luciferase II[20]. BRET probes detecting RAF kinase domain dimerization comprised a C-terminal CAAX-box and were used as previously reported[20]. Flag-tagged ARAF[288–606], BRAF[435–766], CRAF[327–648], KSR1[591–899], and KSR2[644–950] as well as full-length BRAF, CRAF, HRAS, KRAS, and NRAS were generated by cloning the corresponding coding sequences between KpnI and XbaI in a plasmid comprising an N-terminal Flag epitope. Pyo-tagged ARAF[1–287], BRAF[1–434], CRAF[1–326], KSR1[1–558], and KSR2[644–950] fusions were generated by PCR and similarly cloned between KpnI and XbaI. HA-tagged KRAS was inserted between HindIII and BamHI. Point mutations were introduced using the QuikChange II XL site-directed mutagenesis kit (Agilent). All constructs were sequence-verified.

BRAF[444–723] used in TR-FRET experiments was cloned with 16 solubilizing mutations (I543A, I544S, I551K, Q562R, L588N, K630S, F667E, Y673S, A688R, L706S, Q709R, S713E, L716E, S720E, P722S, and K723G)[50], referred to as BRAF16mut, into pPROEX-HTa (Invitrogen) between NcoI and NotI sites.

Small-molecule inhibitors used in this study are listed in Supplementary Data 2. DMSO was systematically used as a vehicle. For BRET and co-IP assays, cells were treated with the indicated concentrations of inhibitor for 2 h at 37 °C. For pERK profiling by AlphaLISA SureFire Ultra assays and Meso Scale Discovery technology, cells were treated for 1 h at 37 °C. DMSO concentration was adjusted to not exceed 0.5%.

**Cell culture and transfections**. HEK293T and HeLa cells were obtained from the IRIC high-throughput screening platform (University of Montreal, Montreal, Canada), which were originally purchased from Sigma-Aldrich (cat. numbers 12022001 and 93021013). HEK293T and HeLa cells were maintained in Dulbecco's Modified Eagle's Medium (DMEM, Sigma) supplemented with 10% heat inactivated fetal bovine serum (FBS, Wisent) at 37 °C under 5% CO$_2$. All cancer cell lines (A375, HCT 116, LOVO, Calu-6, MIA-Paca-2, SW480, Panc10.05, A427, and SW900) were obtained from ATCC and cultured in RPMI-1640 medium (Gibco) supplemented with 10% FBS at 37 °C under 5% CO$_2$. All cell lines are routinely tested for mycoplasma contamination. For HEK293T cell transfections, a standard polyethylenimine (PEI) (PolyScience) transfection protocol was used. Briefly, cells were seeded the day before transfection in DMEM supplemented with 10% FBS at the following densities: 12-well plates: $2.5 \times 10^5$ cells; 100 mm dishes: $2 \times 10^6$ cells; T175 flasks: $5 \times 10^6$ cells. Plasmids were prepared in an appropriate volume of serum-free DMEM and then mixed by gentle vortexing with an equal volume of serum-free DMEM containing PEI (60 μg mL$^{-1}$). Transfection mixes were incubated for 15 min at room temperature and then added to the cells. For BRET assays, cells were assayed 48 h post-transfection, while cells used for co-IPs were lysed 60 h post-transfection.

**BRET titration curves and dose-response curves**. For BRET titration curves, cells were transfected in a 12-well plate format, washed with Hank's Balanced Salt Solution (HBSS; Sigma) and manually re-suspended in 250 μl of HBSS. A volume of 90 μl of cell suspension was then transferred to a white opaque 96-well microtiter plate (BD Biosciences) and incubated with 10 μl of Coelenterazine 400a (2.5 μM final concentration; Gold Biotechnology) for 15 min. BRET signals were read on a Victor Luminescence Counter (PerkinElmer) using BRET2 filters (donor: 400 nm ± 20 nm; acceptor: 510 nm ± 20 nm). For drug dose-response analysis, cells were transfected in 100 mm dishes or T175 flasks and were collected by trypsinization. Cells were rinsed with HBSS once and re-suspended in HBSS at a density of $1 \times 10^6$ cells mL$^{-1}$. A volume of 90 μl of cell suspension was dispensed in a white opaque 96-well microtiter plate and incubated with 10 μl of compound serial dilution for 2 h at 37 °C under 5% CO$_2$. Fifteen minutes before the end of drug treatment, Coelenterazine 400a was added at a final concentration of 2.5 μM for 15 min and BRET2 measurement were made with a Victor Luminescence Counter (PerkinElmer). For both titration experiments and drug dose-response analysis, total GFP$_{10}$ levels were monitored to ensure equal expression of GFP$_{10}$ fusion proteins using a FlexStation II (Molecular Devices) plate reader with excitation and emission peaks set at 400 nm and 510 nm, respectively. BRET2 and GFP$_{10}$ readings were analyzed as described previously[20]. Each BRET experiment was repeated at least three times. BRET curves shown correspond to single representative experiments. Data analysis and curve fitting was done using Graphpad Prism 6.07. BRET titration curves were fitted using a hyperbolic function. Dose-response curves were fitted using a "log (agonist) vs. response—variable slope (four parameters)" function. For all dose-response curves, error bars corresponded to mean values ± s.d. of BRET technical duplicates of a representative biological replicate. Graphpad Prism 6.07 built in "Allosteric EC$_{50}$ shift" analysis[45] was used to determine co-operativity between RAS-GTP and inhibitor action. The ratio between mCherry-KRAS[G12V] (RFU) and GFP10-BRAF (RFU) was used as a proxy for allosteric modulator concentration.

The heatmap presented in Fig. 6b was generated using ClustVis (http://biit.cs.ut.ee/clustvis)[51]. YMAX values and Log$_{10}$-transformed EC$_{50}$s shown in

Supplementary Data 1 were used to hierarchically cluster compounds according to their BRET profile.

**Co-immunoprecipitation and western blotting**. Co-immunoprecipitation and western blotting procedures were done as previously described[20]. Briefly, 48 h post-transfection cells were starved overnight in serum-free DMEM. After appropriate treatment, cells were directly lysed on plates with 1 mL of Triton lysis buffer (50 mM Tris-HCl pH 7.5, 150 mM NaCl, 0.2% Triton X-100, 1 mM EDTA, 10% Glycerol) supplemented with Leupeptin, Aprotinin, PMSF, phosphatase inhibitor cocktail (Sigma) and Na$_3$VO$_4$ at 4 °C for 15 min with gentle rocking. When PPtase λ treatment was required, lysis buffer was supplemented only with Leupeptin, Aprotinin and PMSF to preserve the activity of PPtase λ. Cell lysates were centrifuged at 20,000×g at 4 °C for 10 min. Cleared lysates were then transferred on ice in fresh tubes.

For co-immunoprecipitations, cell lysates were incubated with the appropriate primary antibodies and protein A/G beads (Millipore) with gentle rocking at 4 °C for 4 h. Immunoprecipitates were then washed three times with cold Triton lysis buffer. When indicated, washed beads were incubated with 200 units of PPtase λ (NEB) for 1 h at 30 °C. Otherwise, beads were boiled in 100 μl of 2X sample loading buffer (100 mM Tris-HCl pH 6.8, 4% SDS, 0.2% bromophenol blue, 20% glycerol, 200 mM β-mercaptoethanol) for 5 min prior to SDS-PAGE analysis.

For immunoblotting analysis, whole cell extracts (lysates) or immunoprecipitated proteins were fractionated by SDS-PAGE and transferred to nitrocellulose membranes (PALL). Membranes were blocked for 1 h in TBST (10 mM Tris-HCl pH 8.0, 0.2% Tween-20, 150 mM NaCl) containing 2% BSA (Sigma) and then incubated at 4 °C overnight with a dilution of the following primary antibodies prepared in TBST: anti-BRAF (1:2000; Santa Cruz; cat. number sc-9002), anti-CRAF (1:1000; BD-Millipore; cat. number 610152), anti-Flag (1:5000; Sigma-Aldrich; cat. number F1804), anti-phospho-p44/42 MAPK (1:2000; Sigma-Aldrich; cat. number M9692), anti-total-p44/42 MAPK (1:1000; Cell Signaling Technology; cat. number 4695), anti-panRAS (1:1000; Abcam; cat. number ab108602), anti-phospho-MEK (1:1000; Cell Signaling Technology; cat. number 9121), anti-total-MEK (1:1000; Cell Signaling Technology; cat. number 9122), and anti-KRAS (1:1000; Santa Cruz; cat. number sc-30). HA and Pyo antibodies were produced in-house from hybridoma culture supernatants. Secondary anti-mouse-HRP and anti-rabbit-HRP (Jackson Immunoresearch Labs; cat. number 115-035-146 and 111-035-144, respectively) were prepared in TBST at 1:5000 and 1:10,000 dilutions, respectively. Each co-IP was repeated at least three times. Single representative experiments are shown. Uncropped versions of the most important immunoblots are shown in Supplementary Fig. 10.

**AlphaScreen and meso scale discovery phospho-ERK analysis**. AlphaLISA phospho-ERK analysis was conducted on $4 \times 10^4$ cells cultured overnight in 96-well plates and treated with the indicated compound concentrations for 1 h. AlphaLISA SureFire Ultra pERK 1/2 (Thr202/Tyr204) (PerkinElmer) assays were performed according to the manufacturer's specifications.

For phospho-ERK analysis by Meso Scale Discovery (MSD) technology (electrochemiluminescent ELISA), $2–5 \times 10^4$ cells were seeded overnight in 96-well plates and treated with the indicated compound concentrations for 1 h at 37 °C under 5% CO$_2$. Cells were lysed in MSD pERK lysis buffer, transferred to MSD ELISA plates containing pERK and ERK antibodies (Meso Scale Technologies). Plates were then incubated overnight at 4 °C. The MSD ELISA plates were washed 1X with Tris-buffer saline with 0.5% Tween-20, and then incubated with the sulfotag-labeled ERK antibody for detection. Plates were read on a SECTOR Imager 6000 and phospho-ERK signal was normalized to DMSO control.

The ability of each compound to paradoxically induce ERK signaling in KRAS-mutant cell lines was expressed as maximum pERK signal compared to DMSO controls and was reported alongside the compound concentration at which this maximal signal was observed (Table 1). Each experiment was repeated at least three times. Error bars correspond to mean values ± s.d. of biological triplicates.

**Protein production and purification**. TEV-cleavable WT and R509H 6XHis-tagged BRAF16mut was expressed in *Escherichia coli* BL21(DE3) cells, purified with nickel affinity chromatography, eluted with imidazole and purified through gel filtration chromatography in 20 mM HEPES pH 7.5, 200 mM NaCl, 10 mM DTT and 5% glycerol. Following gel filtration, protein fractions corresponding to greater than 95% purity were pooled and concentrated to 1 mg ml$^{-1}$ (25 μM), aliquoted and then flash frozen in liquid nitrogen.

**Drug-binding assay by TR-FRET**. For drug-binding assays, a procedure similar to the LanthaScreen Eu Kinase Binding Assay for BRAF (Invitrogen) was used. Purified WT or R509H version of 6XHis-tagged BRAF[444–723] kinase domain (50 nM final concentration) were co-incubated with 2 nM LANCE Europium-coupled anti-His antibody (PerkinElmer), 60 nM Alexa Fluor 647-labeled kinase tracer (Invitrogen) and varying concentrations of kinase inhibitors for 1 h at room temperature in kinase buffer (50 mM HEPES pH 7.5, 100 mM NaCl, 3 mM DTT, 10 mM MgCl$_2$, 1 mM EDTA, 0.01% Brij-35). Each experiment included control wells containing the LANCE antibody and Alexa Fluor 647-labeled kinase tracer alone; the average signal of the blank wells was subtracted from each data point.

TR-FRET was read on an Envision (PerkinElmer) plate reader with a $340 \pm 30$ nm excitation filter. The emission of Alexa Fluor 647 signal was monitored with a $665 \pm 10$ nm filter and the Europium emission signal was acquired using a $615 \pm 10$ nm filter. The TR-FRET signal was calculated by dividing the emission signal at 665 nm by the emission at 615 nm. The relative reduction in TR-FRET signal was calculated by normalizing each data point to the DMSO alone-treated wells. Dose-response curves were fitted using a "log (agonist) vs. response—variable slope (four parameters)" function. Each TR-FRET measurement was repeated at least three times. For dose-response curves, error bars corresponded to mean values ± s.d. of TR-FRET technical duplicates of a representative biological replicate.

**Data availability**. All data supporting the findings of the current study are available within the article and its Supplementary Information files or from the corresponding author upon request.

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

## Acknowledgements

We thank P. Beaulieu, C. Baril and J. Gagnon for critical reading of the manuscript. We are grateful to A. Marinier and P. Beaulieu and the IRIC Medicinal Chemistry facility for the synthesis of PLX-0012. HL was supported by a Banting postdoctoral fellowship from

the Canadian Institutes for Health Research (CIHR). This work was supported by an Impact Grant from the Canadian Cancer Society Research Institute (702319) as well as by operating funds from the CIHR (MOP119443) to M.T.

## Author contributions

T.J., H.L. and M.T. designed the experiments and wrote the manuscript. T.J. performed BRET experiments. T.J., M.S. and H.L. performed co-immunoprecipitations. H.L. performed protein production and TR-FRET assays. M.D. and T.J. conducted AlphaLISA SureFire Ultra p-ERK 1/2 (Thr202/Tyr204) experiments and analyses. C.H. and A.H. performed Meso Scale Discovery pERK experiments and analyses.

## Additional information

**Competing interests:** The authors declare no competing financial interests.

