## [Peer Review File · Nature Communications]

Reviewers' comments:

Reviewer #1 (Remarks to the Author):

In this manuscript, Jin et al demonstrated that Raf inhibitors potentiate Ras-Raf interaction through disruption of Raf intramolecular interaction in a manner dependent on Raf dimerization. Further, they also evaluated the effect of newly-developed Raf inhibitors on Raf dimerization, Ras-Raf interaction, and release of BRAf autoinhibition.

Overall, this manuscript is a technically sound and interesting paper. I have several concerns that I think the authors should address prior to publication of the manuscript in its present form.

1. One of my major concerns is the role of Ras-GTP in this scheme.

1-1. Heidorn et al have reported that Ras-GTP is required for Raf dimerization (Cell, 2011). On the other hand, the main claim of this manuscript is that Raf dimerization is required for Ras-Raf complex formation. These two models appear to be inconsistent with each other. The authors should clarify and discuss this discrepancy carefully.

1-2. It has been well established that Ras-GTP binds to CRAf and BRAf to initiate the downstream signaling. However, the second lane in the figure 1d indicates that only a little fraction of Ras-GTP binds to BRAf and CRAf in the absence of Raf inhibitor, whereas third and fourth lanes in the same panel show approximately tens of times higher affinity of Ras-GTP to BRAf and CRAf in the presence of Raf inhibitors. Does this mean only a few percent of BRAf or CRAf is ON state even in overexpression of KRasG12V, and the remaining fraction of Raf is OFF state? This is totally counterintuitive.

2. The other issue is the BRAf autoinhibition by intramolecular bindings.

2-1. BRAf intramolecular FRET biosensor has been reported from Miki Matsuda laboratory, showing that BRAf is homo-dimerized in a head-to-tail dependent manner (Terai, EMBO, 2006). Meanwhile, there is no direct evidence demonstrating the bona fide intramolecular interaction of BRAf NTR and BRAf KD in response to Raf inhibitor in this manuscript. It should be validated by intramolecular BRET system to reveal how Raf inhibitors affect intramolecular binding.

2-2. It is still unclear how Raf inhibitor-bound Raf transactivates a receiver of Raf through dimerization. Is CRAf also autoinhibited by intramolecular binding, and released by Raf inhibitors? Does BRAf NTR bind to CRAf KD?

3. The authors should disclose the quantitative experimental condition, e.g., the stoichiometry of Ras-GTP, BRAf and CRAf. If possible, it is helpful to reveal Kd values for these bindings.

Reviewer #2 (Remarks to the Author):

RAF kinase is a key component of Ras/RAF/MEK/ERK signaling cascade that plays a crucial role in cancer development. Genetic alterations that aberrantly activate Ras/RAF/MEK/ERK signaling exist in over 40% cancers. To target active RAF kinase in cancers, a set of inhibitors have been developed and applied to clinic treatment. Although RAF inhibitors induced objective tumor regression in cancer patients harboring BRAFV600E in a short term, acquired resistance developed rapidly and hampered their clinic efficacy. Mechanistic studies showed that paradoxically activation of RAF kinase by its inhibitors was one of major causes that lead to acquired resistance. Specifically, RAF inhibitors induced dimerization-mediated transactivation of RAF molecules in cancer cells with active Ras, which did not inhibit but promote tumor growth. However, how Raf inhibitors cooperate with active Ras to drive RAF activation remained ambiguous. In this study, Dr. Therrien and his colleague found by using BRET biosensors and biochemistry method that paradoxical activation of RAF kinase was initiated by disruption of intramolecular interaction of NTR-Kinase domain by inhibitor-induced dimerization, which relieved the NTR of RAF molecules and promoted Ras-RAF complex formation. Their data clearly demonstrated that inhibitor-induced dimerization is the first step and the Ras-RAF binding is the second, which was controversial before in this field. This clear-cut study is excellent and reaches the level of Nature Communications. To fulfill requirements of this journal, authors should address questions as follow:

Firstly, in this study, authors showed that RAF ON-state inhibitors disrupted the interaction of NTR- kinase domain of BRAF by using BRET biosensor (Fig3), Does this also happened to two other RAF isoforms (CRAF and ARAF)? Do RAF isoforms have differential affinity of NTR with kinase domain? Does the NTR-kinase domain affinity correlate with the different basal activity of RAF isoforms? Further, does NTR interact with kinase domain in constitutively active cancer mutants such as V600E, G468A?

Secondly, all data in this manuscript clearly showed that NTR docking on kinase domain competed off its dimerization with other kinase domain, though there is no structure of NTR-Kinase domain complex available. If the allostery model is right, RAF inhibitors would not weaken the interaction of NTR with RH kinase domain. Authors might use BIAcore to measure their affinity and further clarify which model is correct. Alternatively, they can examine the overall structure of full-length RAF molecules by Cryo-EM in their future study.

Reviewer #3 (Remarks to the Author):

The manuscript by Jin et al. is a carefully designed study comparing the properties of different classes of RAF inhibitors with the aim of understanding the mechanisms underlying their different behavior in paradox pathway activation. To do this, the authors assess essentially three parameters: i) the effect of the inhibitors on the association of RAF with oncogenic KRAS; ii) the effect of the inhibitors on the intramolecular interaction between the RAF NTR and kinase domain; iii) the effect of the inhibitors on RAF homo- and heterodimerization. These parameters are assessed using BRET sensors and co-ip, mostly of

tagged, ectopically expressed proteins. The study is well designed and overall well conducted, as usual for studies from the Therrien lab.

The major concern with the findings is that most of them are not really novel; in particular, the effect of the inhibitors on RAS/RAF association and on RAF heterodimerization has been recently reported by Karoulia et al. (Cancer Cell 30:485–498, 2016), who came to similar conclusions. The authors cite this paper as ref. 21 of the manuscript; in the discussion, the paper is cited as a demonstration of the fact that “compound-induced RAS-RAF association and relocalization of RAF to the plasma membrane ... remained unexplained”. This sentence doesn’t seem to do justice to the work of Karoulia et al.

The novel finding here is the fact that the inhibitors interfere with the intramolecular interaction of the RAF NTR with the kinase domain. This has never been demonstrated experimentally and is interesting, although considering the impact of the inhibitors on the conformation of the RAF kinase domain it could have been reasonably expected.

Specific comments are listed below:

Fig. 1b and Supplementary Fig. 1b, contrary to what is stated in the text (page 7, lines 118-120), the effect of the compounds on the amplitude is not shown.

Figure 3b and supplementary figure 3b-c, the authors conclude that ERK-dependent phosphorylation of the RAF kinase domain has no impact on its interaction with the kinase domain. However, the co-ip in % Supplementary Figure 3c-d and in Supplementary Figure 4d consistently show that the unphosphorylated RAF kinase domain (lower band) interacts preferentially with the NTR (compare the amount of phosphorylated and unphosphorylated kinase domain present in the cell lysates with that present in the co-ip). This would mean that phosphorylation by presumably ERK prevents, or reduces, intramolecular interaction. Since ERK-induced phosphorylation has been shown to desensitize RAF and to reduce both RAF dimerization and RAS binding, it would seem logical that ERK promotes, not reduces, intramolecular interaction. The authors do not comment on this.

Figure 4d, based on the results obtained using the RAFR509H mutant, the authors conclude that “disruption of RAF intramolecular interaction by inhibitors relies on their ability to promote kinase domain dimerization”. To substantiate the claim, the affinity of the mutant for the inhibitor should be determined.

Supplementary Figure 1c, Figure 5 and Supplementary Figure 5, what is the basis of the different impact of the compounds on i) the binding of BRAF and CRAF to oncogenic RAS; ii) RAF heterodimerization, and on the homodimerization of BRAF vs CRAF? Is there any relationship with the affinity of the inhibitors for the paralogs? What is the effect of the inhibitors on CRAF intramolecular interaction?

Supplementary Figures 1c and 2b, effect of PLX on the KRASG12V-BRAF BRET signal, the EC50 do not match.

Figure 6, in general, it is problematic to group the inhibitors in this way. Examples are: 1) the pathway activating potential of GDC-0879 is very high, while that of AZ628 and LY3009120 is minimal, yet they are in the same “+++” group. 2) Supplementary Figure 4e shows a very high EC50 for AZ628 (low dimer disruption potential, similar to PLX), yet one compound is in the “+++” group. In addition, it is not clear whether the authors have factored in the widely different compound concentrations needed to achieve the effects at study, and if so, how.

Discussion, the authors may consider shortening this section.

On page 17, lines 341-42, the authors discuss the possibility that the NTR might “physically

occlude the side-to-side interface by interacting with specific residues of the interface". If this were the case, these residues would not include R509, since the R509H mutation does not impact interaction with NTR (Figure 4e)

On page 20, lines 394-395, the sentence "Concomitantly, these compounds were shown to inhibit kinase domain dimerization as measured in vitro with purified protein" is misleading - the inhibitors promote RAF dimerization in vivo, albeit at a lower level than the type I/II inhibitors.

Responses to reviewers' *verbatim* comments

Reviewer 1

In this manuscript, Jin et al demonstrated that Raf inhibitors potentiate Ras-Raf interaction through disruption of Raf intramolecular interaction in a manner dependent on Raf dimerization. Further, they also evaluated the effect of newly-developed Raf inhibitors on Raf dimerization, Ras-Raf interaction, and release of BRaf autoinhibition.

Overall, this manuscript is a technically sound and interesting paper.

We thank the reviewer for acknowledging that our manuscript is technically sound and interesting.

I have several concerns that I think the authors should address prior to publication of the manuscript in its present form.

1. One of my major concerns is the role of Ras-GTP in this scheme.

1-1. Heidorn et al have reported that Ras-GTP is required for Raf dimerization (Cell, 2011). On the other hand, the main claim of this manuscript is that Raf dimerization is required for Ras-Raf complex formation. These two models appear to be inconsistent with each other. The authors should clarify and discuss this discrepancy carefully.

It is indeed well accepted that GTP-loaded RAS is a physiological trigger of RAF dimerization and activation. RAS-GTP is also required for the ability of RAF inhibitors to promote RAF dimerization. These properties of RAS-GTP have been demonstrated by a number of groups in addition to Heidorn *et al.*¹⁻³, and they are not questioned here. Yet, RAF inhibitors do enhance RAS-RAF complex formation, but the mechanism is not known. This is specifically the question we are addressing in this study. We found that this phenomenon is in part linked to the ability of RAF inhibitors to promote kinase domain dimerization. Interestingly, we also found that RAF inhibitors disrupt the NTR-KD interaction independently of RAS activity. By altering the NTR-KD interaction, RAF inhibitors may facilitate the access of RAS-GTP to the RBD.

Nevertheless, it remains that RAF inhibitors barely promote the dimerization of endogenous RAF proteins without RAS activity. This could be due to the fact that RAS serves as a nucleating center through its ability to form nanoclusters at the plasma membrane^{4,5}. The clustering of RAF in a bi-dimensional space conferred by the plasma membrane increases the effective concentration of RAF molecules and thereby promotes dimer formation. Conversely, it has been shown that RAS nanoclustering is further reinforced by RAF inhibitors⁶.

Considering that RAS activity and RAF inhibitors impinge on related events that concur to promote RAS-RAF complex formation and RAF dimerization, and that their combination produces synergistic effects, it suggests that they act cooperatively. We now demonstrate this point in our revised manuscript (Fig. 4 and Suppl. Fig. 7).

To clarify the respective role of RAS activity and RAF inhibitors, we present a revised model through four vignettes that detail how RAF inhibitors enhance RAS-RAF complex formation and RAF dimerization (revised Fig. 6a). This model highlights the cooperative nature of the distinct events at play.

1- Without RAS activity and without RAF inhibitors: In the absence of RAS activity, RAF is maintained inhibited in the cytoplasm through an intramolecular interaction between its kinase domain and the N-terminal regulatory region (NTR).

- 2- RAS activity only: RAS-GTP associates with the RAF RBD at a certain rate, which on the one hand, disrupts the NTR-KD interaction, and on the other hand, promotes RAF kinase domain dimerization owing to RAS nanoclustering property (increases RAF effective concentration at the plasma membrane).
- 3- RAF inhibitor only: RAF inhibitor binding lead to a close/active-like state of the RAF kinase domain ⁷, which is conducive to kinase domain dimerization. Binding also leads to disruption of the NTR-KD interaction. However, the affinity of the side-to-side dimer interface of the RAF kinase domain is not sufficiently high to support the formation of productive dimers in the cytoplasm, thus resulting in negligible dimer formation and catalytic activation. In addition, the absence of RAS binding to the NTR might allow a certain rate of NTR-KD re-association and thereby further weaken dimer formation.
- 4- With RAS activity and RAF inhibitors: By acting at distinct levels, both RAS activity and RAF inhibitors cooperatively promote RAS-RAF complex formation and RAF dimerization. On the one hand, NTR-KD disruption is triggered by two separate means (RAS-binding to the RBD and compound binding to the catalytic cleft) thereby resulting in effective disruption. On the other hand, the increase in RAF effective concentration when bound to RAS nanoclusters coupled with the active conformation stabilized by the inhibitors leads to an increase in side-to-side RAF dimerization. In turn, RAF dimerization further increases the population of RAS nanoclusters. Together, these events cooperatively drive RAS-RAF complex formation and RAF dimerization.

1-2. It has been well established that Ras-GTP binds to CRaf and BRaf to initiate the downstream signaling. However, the second lane in the figure 1d indicates that only a little fraction of Ras-GTP binds to BRaf and CRaf in the absence of Raf inhibitor, whereas third and fourth lanes in the same panel show approximately tens of times higher affinity of Ras-GTP to BRaf and CRaf in the presence of Raf inhibitors. Does this mean only a few percent of BRaf or CRaf is ON state even in overexpression of KRasG12V, and the remaining fraction of Raf is OFF state? This is totally counterintuitive.

No studies have specifically addressed the stoichiometry of endogenous RAF bound to constitutively activated KRas at steady state levels. Therefore, there is no reference for comparison. We agree that the enhanced RAS-RAF interaction upon drug treatment looks impressive, but this is what we reproducibly observed using various inhibitors, albeit some compounds show significantly lower inductions as detailed in the manuscript (e.g., Fig. 1d, lanes 6-7). It is important to note that the non-induced signal (lane 2) may look faint, but this is due to the short film exposure used in order to visually appreciate the dynamic range of the inductions.

We systematically used 400 ng of the Flag-KRas^{G12V} construct in our experiments, which enabled the detection of a reproducible and non-saturated signal upon drug treatment. Indeed, we determined this amount by assessing the induction signal of a range of plasmid quantities (50, 100, 200, 400, 800 ng) and found a reproducible induction of ~ 12-fold for BRAF and ~8-fold for CRAF with the 100-400 ng range of the transfected KRas^{G12V} construct (revised Suppl. Fig. 2a) (quantified by Image J using non-saturated exposures). Lower plasmid quantities gave variable inductions since the unstimulated condition produced signals too close to background. Conversely, the induced signals at higher plasmid quantities (800 ng) tended to be dampened owing to rapid saturation of the film.

The induced interactions upon drug treatment were not specific to the G12V allele or to the KRas protein. Indeed, we made similar observations with another KRas oncogenic allele (Q61H; revised Suppl. Fig 2b) as well as with HRas^{G12V} or NRas^{G12V} (Fig. 1d, right panel). Taken together, these results suggest an increase in the affinity of the RAS-RAF interaction upon inhibitor treatment, which resides (as

detailed in point 1.1 above) in the ability of the compound to release the intramolecular interaction and to stabilize RAF kinase domain dimers at the plasma membrane.

2. *The other issue is the BRAf autoinhibition by intramolecular bindings.*

2-1. *BRAf intramolecular FRET biosensor has been reported from Miki Matsuda laboratory, showing that BRAf is homo-dimerized in a head-to-tail dependent manner (Terai, EMBO, 2006). Meanwhile, there is no direct evidence demonstrating the bona fide intramolecular interaction of BRAf NTR and BRAf KD in response to Raf inhibitor in this manuscript. It should be validated by intramolecular BRET system to reveal how Raf inhibitors affect intramolecular binding.*

This is an interesting point raised by the reviewer. We also reasoned that intramolecular BRET probes could be useful to support our model. We built such biosensors with the intention of including them in the manuscript. We generated both GFP10-BRAF-RlucII and RlucII-BRAF-GFP10 intramolecular probes comprising WT BRAF or BRAF^{R509H} and tested the effect of activated KRAS and of GDC-0879 on the BRET signals. However, these experiments were inconclusive for the following reasons:

- 1- Firstly, intramolecular BRET probes cannot discriminate intramolecular from inter-molecular interactions. In fact, it appears that the release of autoinhibition could not be distinguished from dimerization and plasma membrane nanoclustering in the context of these dual BRET fusions:
 - a. Titrating Ras led to an increase in BRET signal rather than the expected decrease that one would predict for probes reporting on a release of autoinhibition. We note that Terai and co-workers (2006) similarly detected an increase in FRET signal for their YFP-BRAF-CFP (Prin-BRAF) construct upon Ras stimulation with EGF (Terai *et al.* 2006, Fig. 1c).
 - b. We observed that GDC-0879, LY-3009120 and PLX4032 induced BRET signals and this, regardless of *bona fide* side-to-side dimerization (the R509H mutation did not alter the response). We therefore could not rely on the biosensor to monitor genuine kinase domain dimerization and therefore they are also useless to monitor the NTR-KD interaction.
- 2- Secondly, we observed that fusion of the GFP10 and RlucII moieties at the N- and C-terminus of RAF proteins (in both orientations) interferes with function. Indeed, we found that both GFP10-BRAF-RlucII and RLucII-BRAF-GFP10 constructs led to constitutive kinase activation, which is never seen with singly tagged constructs. Furthermore, this elevated catalytic activity was not blocked by the R509H mutation, indicating that the constructs did not behave like WT RAF. These results imply that the probes 1) interfered with normal autoinhibition and 2) caused dimerization-independent activation of BRAF.

These results led us to conclude that the BRET probes described above were inappropriate for studying RAF intramolecular interaction and thus significant amount of work would still need to be invested in order to develop reliable tools for addressing this question in a full-length context.

It is important to note that FRET or BRET signal modulations might reflect conformational changes rather than gain or loss of protein-protein interaction and therefore they need to be systematically validated by biochemical means. For instance, intramolecular beta-arrestin BRET biosensors (detect GPCR association) behave oppositely depending on whether the YFP and luciferase moieties are located at the N- or C-terminal extremity^{8,9}.

2-2. *It is still unclear how Raf inhibitor-bound Raf transactivates a receiver of Raf through dimerization. Is CRAf also autoinhibited by intramolecular binding, and released by Raf inhibitors? Does BRAf NTR bind to CRAf KD?*

With respect to the first question, CRAF had been previously shown (like BRAF) to be autoinhibited by intramolecular binding^{10,11}. However, the release of interaction by inhibitors has not been determined.

This is an important point raised by the reviewer. As we were setting out to verify this question, we reasoned that it would be informative to systematically determine the ability of each RAF family members to undergo intramolecular binding and then test their response to RAF inhibitors. We thus generated the appropriate constructs and tested these by co-IP. Interestingly, in addition to BRAF and CRAF, we detected NTR-KD associations for the three other members (ARAF, KSR1, and KSR2; revised Fig. 4a), which had not been previously documented. These findings support the notion that intramolecular binding is a common feature of RAF family members. We also noticed significant differences in interaction strength, suggesting differential affinities between RAF family members, which may be relevant for their regulatory mode. In decreasing order, interaction strength is as follows: KSR1 > CRAF > KSR2 > BRAF > ARAF (revised Fig. 4a).

Next, we assessed how each NTR-KD interaction responded to type I and type II inhibitor treatment (revised Fig. 4b). Interestingly, in addition to BRAF, only the CRAF NTR-KD interaction was disrupted in response to drug treatment, whereas ARAF, KSR1 and KSR2 showed no response.

As for the second question (Does BRAf NTR bind to CRAf KD?), we note that it had previously been addressed by Tran *et al.* (2005), although in the other orientation. Indeed, they showed that the CRAF NTR can interact with and inhibit the activity of BRAF KD (Fig. 5a-b of Tran *et al.* 2005)¹². Despite this evidence, we went ahead and compared the capacity of the BRAF and CRAF NTRs to associate with the CRAF KD. Interestingly, both NTRs had indeed a similar ability to associate with the CRAF KD (**Fig. A**). Furthermore, both of these interactions were impaired by GDC-0879, indicating again that the allosteric influence of compounds on NTR-KD complexes is conserved between paralogs. Taken together, these results suggest that the BRAF-CRAF NTRs and kinase domains can interact in *trans* in addition to their *cis*-interactions. Whether this occurs under physiological conditions remains to be determined. Be that as it may, it is important to note that in full-length proteins, the covalent link that bridges intramolecularly interacting domains (such as the RAF NTR and KD) boosts their effective concentration to the millimolar range (reviewed in Kuryian and Eisenberg^{13,14}). This greatly exceeds the usual concentration of isolated proteins in cells (nM range). We therefore do not think that this *trans*-interaction mode competes away the *cis* interaction (unless other regulatory events took place). As this does not represent a central point in our manuscript and because our current version is relatively long, we decided not to include this new information. However, if the reviewer judges that this should be included, we will do so.

Figure A

3. The authors should disclose the quantitative experimental condition, e.g., the stoichiometry of Ras-GTP, BRAf and CRAf. If possible, it is helpful to reveal Kd values for these bindings.

We agree with the reviewer that it would be informative to derive a quantitative assessment of the different interactions described in this study. However, this would involve non-trivial and time-consuming experiments that have hardly been addressed by any group in the field investigating similar questions.

While this would allow a quantitative appreciation of the various association / dissociation at play, the information would not change the main conclusions drawn from our work. Moreover, as our experiments were performed in cells and not in reconstituted systems, the role of peripheral partners such as scaffolds, 14-3-3 proteins, lipids, etc. is currently not appreciated and their absence in reconstituted systems would likely complicate Kd determination. Therefore, we respectfully argue that this request goes beyond the scope of the current study.

Reviewer 2

RAF kinase is a key component of Ras/RAF/MEK/ERK signaling cascade that plays a crucial role in cancer development. Genetic alterations that aberrantly activate Ras/RAF/MEK/ERK signaling exist in over 40% cancers. To target active RAF kinase in cancers, a set of inhibitors have been developed and applied to clinic treatment. Although RAF inhibitors induced objective tumor regression in cancer patients harboring BRAFV600E in a short term, acquired resistance developed rapidly and hampered their clinic efficacy. Mechanistic studies showed that paradoxically activation of RAF kinase by its inhibitors was one of major causes that lead to acquired resistance. Specifically, RAF inhibitors induced dimerization-mediated transactivation of RAF molecules in cancer cells with active Ras, which did not inhibit but promote tumor growth. However, how Raf inhibitors cooperate with active Ras to drive RAF activation remained ambiguous. In this study, Dr. Therrien and his colleague found by using BRET biosensors and biochemistry method that paradoxical activation of RAF kinase was initiated by disruption of intramolecular interaction of NTR-Kinase domain by inhibitor-induced dimerization, which relieved the NTR of RAF molecules and promoted Ras-RAF complex formation. Their data clearly demonstrated that inhibitor-induced dimerization is the first step and the Ras-RAF binding is the second, which was controversial before in this field. This clear-cut study is excellent and reaches the level of Nature Communications. To fulfill requirements of this journal, authors should address questions as follow:

We thank the reviewer for the kind comments on our work.

1- Firstly, in this study, authors showed that RAF ON-state inhibitors disrupted the interaction of NTR-kinase domain of BRAF by using BRET biosensor (Fig3), Does this also happened to two other RAF isoforms (CRAF and ARAF)?

This is an interesting question that is related to those formulated by Reviewers 1 and 3. As presented above to answer Reviewer 1; Point 2.1, we have now addressed the ability of RAF inhibitors to impede the NTR-KD interaction of each RAF family member. In addition to BRAF, we discovered that the CRAF NTR-KD interaction was also disrupted by inhibitors. However, the NTR-KD interaction also observed for ARAF, KSR1, and KSR2 were unresponsive to drug treatment (revised Fig. 4b).

2- Do RAF isoforms have differential affinity of NTR with kinase domain?

We expressed similar amounts of the NTR and KD of each RAF family member in 293T cells and compared the ability of each pair to interact by co-IP. This allowed two interesting observations (revised Fig. 4a). Firstly, it showed that each member can indeed undergo intramolecular binding (besides BRAF and CRAF, this was not known for the other proteins). Secondly, it suggested differential affinity between the distinct pairs. In decreasing order, interaction strength showed the following trend: KSR1 > CRAF > KSR2 > BRAF > ARAF. These new results are presented in the revised manuscript (lines 310 to 324).

3- Does the NTR-kinase domain affinity correlate with the different basal activity of RAF isoforms?

The strength of the NTR-KD interactions had no apparent correlation with the propensity of each kinase domain to phosphorylate MEK. This being said, besides BRAF and CRAF, ARAF, KSR1, and KSR2 did not demonstrate phospho-transfer activity towards the MEK activation loop (revised Suppl. Fig. 6). Thus, if we only considered BRAF and CRAF as catalytically competent enzymes, it is interesting to note that BRAF has an apparently weaker intramolecular binding affinity than CRAF, which correlates with its higher basal kinase activity¹⁵. We briefly mention these new observations in the Results section (lines 315-317).

4- *Further, does NTR interact with kinase domain in constitutively active cancer mutants such as V600E, G468A?*

To address this question, we performed co-IP in 293T cells to assess the NTR-KD interaction of various BRAF oncogenic mutants (T599I, V600E, insT, TV->IAL, G464V, G469V and N581S; revised Suppl. Fig. 5b). Interestingly, each mutant kinase domain interacted with the BRAF NTR with a propensity similar to that of the WT BRAF kinase domain. Together, these results are consistent with previous publications by Tran *et al.*¹² and Chong *et al.*¹⁰ who had documented similar interactions using respectively the BRAF_V600E or CRAF_DEDD hyperactivated variants.

Below is the relevant citation from Tran *et al.* 2005¹²:

“As shown in Fig. 9A, although ERK2 activation by wild-type B-RafCAT was progressively inhibited by expression of B-Raf-(1–435), ERK2 activity stimulated by B-RafCAT V599E - (now known as V600E) - was unaffected by expression of the B-Raf N terminus. These data indicate that the V599E mutation blocks autoinhibitory activity. To test whether this mutation also affects the ability of the N terminus to interact with the catalytic domain, we examined the ability of B-Raf-(1–435) to coprecipitate with B-RafCAT V599E. As shown in Fig. 9B, the B-Raf N terminus coprecipitated with B-RafCAT V599E as efficiently as with wild-type B-RafCAT. These findings suggest that, similar to the T598E/S601D substitutions, the V599E mutation blocks autoinhibition by affecting the manner in which the autoinhibitory and catalytic domains interact, rather than precluding their binding altogether.”

Chong and colleagues came to the same conclusion using a CRAF_DDED mutant (338D/Y341D/T491E/S494D), which comprises negatively-charged region and activation segment mutations mimicking the BRAF_V600E variant. These studies and our results indicate the equal ability of the BRAF NTR to interact with WT or oncogenically activated BRAF kinase domains. However, as described in the two previous studies, the interaction does not necessarily translate in autoinhibition with the oncogenic mutants, although later work hinted that the NTR can still suppress BRAF V600E kinase activity to some extent³.

Next, we tested whether the NTR-KD interaction of the V600E and G469V oncogenic alleles could still be disrupted by GDC-0879 treatment. Similar to WT BRAF, the NTR-KD interaction of V600E and G469V variants was impaired upon compound treatment (revised Suppl. Fig. 5c). This indicates that the mechanism that we propose operates not only in WT BRAF cells, but also in BRAF-mutant cells.

It is interesting to note that even though these oncogenic kinase domain variants appeared to interact normally with their NTR, full-length versions of these mutants systematically interacted stronger with RAS-GTP (revised Suppl. Figs. 5a-b). This suggests that oncogenic mutations do not generally reduce the affinity of the NTR-KD interaction, but possibly conformationally alter the interaction in a way that increases RBD access to RAS-GTP. These new findings are presented in the revised version (lines 301 to 307).

5- *Secondly, all data in this manuscript clearly showed that NTR docking on kinase domain competed off its dimerization with other kinase domain, though there is no structure of NTR-Kinase domain complex available. If the allosteric model is right, RAF inhibitors would not weaken the interaction of NTR with RH kinase domain.*

This is an important point that we attempted to address in our initial submission. Indeed, in the original Fig. 4e, we showed that the BRAF NTR-KD_R509H interaction was not disrupted by GDC-0879 treatment. We used this result as an argument to suggest that dimerization plays a role in compound-induced NTR-KD disruption. However, as judiciously pointed out by Reviewer 3 (see Reviewer 3, point

#4 below), this argument is valid as long as BRAF_R509H binds GDC-0879 with the same affinity as WT BRAF.

We therefore set out to address this important point. For this, we produced and purified WT and R509H versions of the BRAF kinase domain from bacteria and measured GDC-0879 binding affinity using a time-resolved FRET (TR-FRET) assay as previously described⁷. Unexpectedly, we observed a 7-fold reduction for the R509H variant compared to WT (revised Fig. 2d). We concluded that this loss of affinity at least partly explained the reduced potency to drive RAS-RAF interaction (original Figs. 2c-d) as well as its reduced ability of GDC-0879 to disrupt the NTR-KD_R509H interaction (original Fig. 4e). We therefore decided to compare the affinity of another Type I inhibitor (SB590885) as well as three Type II inhibitors (AZ-628, LY-3009120, and dabrafenib). Interestingly, SB590885 also showed a 4-fold loss of affinity, whereas the three Type II inhibitors showed equal affinity for WT and R509H BRAF (revised Fig. 2d). These data suggested that the binding affinity of Type I inhibitors is affected by the R509H mutation, whereas Type II inhibitors are not. This is an important new piece of information that no one had suspected and which will need to be taken into account in future studies using Type I inhibitors and their impact on RAF dimerization.

As Type II inhibitors were not affected, we decided to revisit the impact of the R509H mutation on the compounds' ability to promote RAS-RAF interaction. Instead of a strong loss of potency observed with Type I inhibitors, we saw a modest 2-fold drop with Type II inhibitors (revised Fig. 2c). This correlated with a 2 to 4-fold reduction in the ability of Type II inhibitors to drive the dimerization of BRAF_R509H (revised Fig. 2e). Although this suggests that dimerization plays a role in RAS-RAF interaction, the fact that the R509H mutation does not strongly impede dimerization induced by Type II inhibitors prevented us from determining whether dimerization is critical or simply one component among others in this event.

We then went on and tested the ability of Type II inhibitors to disrupt the NTR-KD_R509H interaction. Unexpectedly, the mutation hardly had any effect (revised Fig. 3g). Taken at face value, these new findings suggest that dimerization *per se* plays a minor role if any in the ability of RAF inhibitors to disrupt the NTR-KD interaction. This conclusion, however, has to be taken cautiously since Type II inhibitors can still induce, albeit with a 2 to 4-fold reduced potency, the homodimerization of BRAF_R509H (revised Fig. 2e). These new observations led us to propose that compound occupancy of the kinase domain cleft and its specific effect on the kinase domain conformation likely alter in *cis* the NTR-KD interaction and, in turn, physically abrogates the interaction or modifies it in such a way that it positively influences the accessibility of the RAF RBD for RAS-GTP. The new results are part of two completely reworked sections investigating compound-induced RAS-RAF interaction (revised Fig. 2 and Suppl. Fig. 3) (lines 182 to 225) and compound-induced NTR-KD disruption (revised Fig. 3 and Suppl. Figs. 4-5) (lines 227 to 307).

6- Authors might use BLAcore to measure their affinity and further clarify which model is correct. Alternatively, they can examine the overall structure of full-length RAF molecules by Cryo-EM in their future study.

We agree with the reviewer that more quantitative and structure-based experiments would be helpful. However, these approaches require sizable amounts of purified proteins and, despite multiple attempts, we and others are still unable to produce enough of the BRAF NTR to conduct reliable binding experiments. Likewise, structural studies capturing the autoinhibited state or the dimer-activated state of full-length RAF proteins are still lacking mainly due to technical reasons linked to the enormous challenge of producing soluble and well-behaved proteins. Thus far, successful protein production has been limited to the isolated kinase domain (mainly BRAF), and two separate NTR domains (e.g. RBD and CRD). Newer approaches, such as cryo-EM-based methods may eventually solve the problem (at least for RAF-containing complexes larger than 150 KDa, which is the minimal size that cryo-EM can

resolve for now), but we are not there yet as producing such protein complexes in sufficient amount and purity has not been achieved, and remains a major hurdle.

Reviewer 3

The manuscript by Jin et al. is a carefully designed study comparing the properties of different classes of RAF inhibitors with the aim of understanding the mechanisms underlying their different behavior in paradox pathway activation. To do this, the authors assess essentially three parameters: i) the effect of the inhibitors on the association of RAF with oncogenic KRAS; ii) the effect of the inhibitors on the intramolecular interaction between the RAF NTR and kinase domain; iii) the effect of the inhibitors on RAF homo- and heterodimerization. These parameters are assessed using BRET sensors and co-ip, mostly of tagged, ectopically expressed proteins. The study is well designed and overall well conducted, as usual for studies from the Therrien lab.

We are grateful to the reviewer for qualifying our work as well designed and well conducted.

1- The major concern with the findings is that most of them are not really novel; in particular, the effect of the inhibitors on RAS/RAF association and on RAF heterodimerization has been recently reported by Karoulia et al. (Cancer Cell 30:485–498, 2016), who came to similar conclusions. The author cite this paper as ref. 21 of the manuscript; in the discussion, the paper is cited as a demonstration of the fact that “compound-induced RAS-RAF association and relocalization of RAF to the plasma membrane ... remained unexplained”. This sentence doesn’t seem to do justice to the work of Karoulia et al. The novel finding here is the fact that the inhibitors interfere with the intramolecular interaction of the RAF NTR with the kinase domain. This has never been demonstrated experimentally and is interesting, although considering the impact of the inhibitors on the conformation of the RAF kinase domain it could have been reasonably expected.

The effects of inhibitors on Ras-RAF association and RAF dimerization has been studied prior to Karoulia *et al.* (2016)¹⁶ by several labs including our own. Together, these studies brought to light the allosteric impact of inhibitors in driving RAF dimerization and transactivation. The structural details, however, still remain elusive. Karoulia *et al.* shed some light on this front and their contribution is captured by the following sentence in their discussion: “*Here we show that inhibitor-induced RAF dimerization is downstream of RAF binding to RAS-GTP as a result of a movement around the R506 residue in the α C helix of BRAF. This suggests that this area may be critical in the interaction of the catalytic domain with the N terminus of RAF. Unfortunately, the lack of crystallographic data on RAF encompassing its N terminus currently precludes further structural validation of the role of R506.*”

Their suggestion that the conformation of the R506 side chain modulates the NTR-KD interaction is however based solely on correlative evidence. They actually did not test the effect of R506 substitutions on the NTR-KD interaction nor on the RAS-RAF interaction. Therefore, the mechanism responsible for compound-induced RAS-RAF association remains to be worked out, which is what we meant by the specific sentence referred to by the reviewer. We agree that we could have better acknowledged this work. We therefore added the following text in the discussion section (line 420) that recognizes their contribution:

“Karoulia et al. (2016) recently reported that RAF dimerization induced by inhibitors requires RAS-GTP binding to RAF. Interestingly, they correlated the propensity of inhibitors to induce dimerization and RAS-RAF binding to the compounds’ ability to re-orient the side chain of the R506 residue situated nearby the

dimerization interface at the C-terminal end of helix α C. They suggested that this area might play a role in the NTR-KD interaction, however, this remains to be experimentally investigated.”

With respect to the novelty of our work, we discern at least two novel contributions that, we argue, represent significant advances to the field. Firstly, we show for the first time that ON-state RAF inhibitors disrupt the BRAF NTR-KD interaction (we now show that this also holds true for CRAF). We demonstrate that this occurs independently of RAS activity and show that it partly correlates with the compounds' ability to drive kinase domain dimerization. These new findings suggest that other event(s) also enable inhibitors to alter the NTR-KD interaction. In our revised version, we also show that RAS-GTP and RAF inhibitors work in a cooperative manner owing to their distinct, but converging effects on RAF conformation. Our work thus identifies the NTR-KD interaction as a new and useful parameter for RAF inhibitor development.

Secondly, we now show that all RAF family members undergo NTR-KD interaction and exhibit distinct affinities. Although it is assumed that RAF proteins are regulated via autoinhibition implemented by NTR-KD interaction, this has remained an enigmatic phenomenon left on the “back burner” given the lack of structural information. As this appears to be a general phenomenon among RAF family members that can be modulated by small molecules, we argue that our work will contribute to put this event on the center stage and stimulates interest for a detailed structural characterization. Such information should prove highly valuable for designing new classes of small molecules that stabilize the BRAF and/or CRAF NTR-KD interaction and thereby lead to genuine OFF-state inhibitors.

Specific comments are listed below:

2- Fig. 1b and Supplementary Fig. 1b, contrary to what is stated in the text (page 7, lines 118-120), the effect of the compounds on the amplitude is not shown.

The reviewer is correct. In fact, this comment only applies to PLX4032. We added the YMAX values calculated for each curve in Supp. Table 1. The lines referring to this comment are 143-145 in the revised text.

3- Figure 3b and supplementary figure 3b-c, the authors conclude that ERK-dependent phosphorylation of the RAF kinase domain has no impact on its interaction with the kinase domain. However, the co-ip in % Supplementary Figure 3c-d and in Supplementary Figure 4d consistently show that the unphosphorylated RAF kinase domain (lower band) interacts preferentially with the NTR (compare the amount of phosphorylated and unphosphorylated kinase domain present in the cell lysates with that present in the co-ip). This would mean that phosphorylation by presumably ERK prevents, or reduces, intramolecular interaction. Since ERK-induced phosphorylation has been shown to desensitize RAF and to reduce both RAF dimerization and RAS binding, it would seem logical that ERK promotes, not reduces, intramolecular interaction. The authors do not comment on this.

This is an interesting observation and a relevant remark made by the reviewer, but for which we have no definitive answer. The robust phosphorylation of the BRAF kinase domain upon GDC-0879 treatment appears to be ERK-dependent since it is abrogated by the ERK inhibitor SCH772984 (revised Suppl. Fig. 4d). We have not mapped the phosphorylation site(s) and thus do not know whether they only comprise ERK phospho-sites or also include phospho-sites mediated by other kinases (which could have differential impact). We note that the isolated BRAF kinase domain does not significantly shift when expressed without GDC-0879 even though ERK is robustly activated under this condition (Fig. 3c, anti-Flag panel, for example). This indicates that GDC-0879 binding likely induces a conformational change in the kinase domain that renders it susceptible for ERK-dependent phosphorylation (this does not necessarily mean that ERK is the responsible kinase as another ERK-induced kinase could be directly involved). Importantly, since GDC-0879 treatment disrupted NTR-KD interaction even in the presence

of the ERK inhibitor (revised Suppl. Fig. 4d), it confirmed that NTR-KD disruption by GDC-0879 can proceed independently of ERK activity.

The mobility shift of the BRAF kinase domain was confined to the GDC-0879 compound as none of the other inhibitors tested had a similar effect. For instance, LY-3009120 disrupted NTR-KD complexes without altering the phosphorylation pattern of BRAF or CRAF kinase domains (Fig. 4b and Fig. 5g). Furthermore, the mobility of the CRAF KD was not strongly affected by GDC-0879, yet it was compromised in its ability to associate with its own NTR upon GDC-0879 treatment (revised Fig. 4b). Together, our data provide compelling evidence that NTR-KD disruption by inhibitors does not require ERK activity.

We agree that a simple model would be that ERK negative feedback phosphorylation would not only reduce RAS-RAF interaction and RAF dimerization, but also augment RAF intramolecular binding. However, our comprehension of the RAF activity cycle is still fragmented and it could well be that following ERK phosphorylation, the NTR plays no immediate role in inhibiting the kinase domain (kinase domain phosphorylation might suffice to inhibit kinase activity). Previous work from the Morrison lab have shown that following ERK-mediated phosphorylation, CRAF needs to be dephosphorylated by PP2A in a Pin1-dependent manner¹⁷. Only after this dephosphorylation step did CRAF regain its “signal-activatable” competence.

While speculative at this stage, it is possible that it is only upon kinase domain dephosphorylation that the NTR re-associates with the kinase domain in order to keep the kinase activity in check until a new round of upstream signals comes in. Significant work would be required to tease out the putative impact of kinase domain phosphorylation on the NTR interaction, in the presence or absence of inhibitors. Although interesting, this question is peripheral to our current study and thus was not further investigated. However, we took note of the reviewer’s advice and introduced a comment to this effect in the revised manuscript (lines 256-261).

4- Figure 4d, based on the results obtained using the RAFR509H mutant, the authors conclude that “disruption of RAF intramolecular interaction by inhibitors relies on their ability to promote kinase domain dimerization”. To substantiate the claim, the affinity of the mutant for the inhibitor should be determined.

This is an important point raised by the reviewer and we are grateful that it has been asked as it enabled us to discover that indeed GDC-0879 binding to BRAF was impeded by the R509H mutation and thus could not be reliably used. The answer presented hereafter is the same as the one provided above for Reviewer 2 (point #5) who asked a related question.

To address the issue, we produced and purified WT and R509H versions of the BRAF kinase domain from bacteria and measured GDC-0879 binding affinity using a time-resolved FRET (TR-FRET) assay as previously conducted⁷. Unexpectedly, we observed a 7-fold reduction for the R509H variant compared to WT (revised Fig. 2d). We concluded that this loss of affinity likely explained the inability of GDC-0879 to disrupt the NTR-KD_R509H interaction (original Fig. 4e) as well as its reduced potency to drive RAS-RAF interaction (original Figs. 2c-d). We therefore decided to compare the affinity of another Type I inhibitor (SB590885) as well as three Type II inhibitors (AZ-628, LY-3009120, and dabrafenib). Interestingly, SB590885 also showed a 4-fold loss of affinity, whereas the three Type II inhibitors showed normal affinity for the BRAF_R509H kinase domain (revised Fig. 2d). These data suggested that the binding affinity of Type I inhibitors is affected by the R509H mutation, whereas Type II inhibitors are not. This is an important new piece of information that no one had suspected and which will need to be taken into account in future studies on Type I inhibitors and their impact on RAF dimerization.

As Type II inhibitors were not affected, we decided to revisit the impact of the R509H mutation on the compounds’ ability to promote RAS-RAF interaction. Instead of a strong loss of potency observed with Type I inhibitors, we saw a modest 2-fold drop with Type II inhibitors (revised Fig. 2c). This

correlated with a 2 to 4-fold reduction in the ability of Type II inhibitors to drive the dimerization of BRAF_R509H. Although this suggests that dimerization plays a role in RAS-RAF interaction, the fact that the R509H mutation does not strongly impede dimerization induced by Type II inhibitors, prevents us from determining whether dimerization is critical or simply one component among others in this event.

We then went on and tested the ability of Type II inhibitors to disrupt the NTR-KD_R509H interaction. Unexpectedly, the mutation hardly had any effect (revised Fig. 3g). Taken at face value, these new findings suggest that dimerization *per se* plays a minor role if any in the ability of RAF inhibitors to disrupt the NTR-KD interaction. This conclusion, however, has to be taken cautiously since Type II inhibitors can still induce, albeit with a 2 to 4-fold reduced potency, the dimerization of BRAF_R509H (Fig. 2e). These new observations led us to propose that compound occupancy of the kinase domain cleft and its specific effect on the kinase domain conformation likely alter *in cis* the NTR-KD interaction and, in turn, physically abrogates the interaction or modifies it in such a way that it positively influences the accessibility of the RAF RBD for RAS-GTP. The new results are part of two completely reworked sections investigating compound-induced RAS-RAF interaction (Fig. 2 and Suppl. Fig. 3) (lines 182 to 225) and compound-induced NTR-KD disruption (revised Fig. 3 and Suppl. Figs. 4-5) (lines 227 to 307).

5- Supplementary Figure 1c, Figure 5 and Supplementary Figure 5, what is the basis of the different impact of the compounds on i) the binding of BRAF and CRAF to oncogenic RAS; ii) RAF heterodimerization, and on the homodimerization of BRAF vs CRAF? Is there any relationship with the affinity of the inhibitors for the paralogs?

Binding affinity plays a major role in the distinctive patterns of compound-induced RAF dimerization. This was previously addressed in Lavoie *et al.* (2013), where we established that a strong correlation exists between binding IC₅₀'s (TR-FRET) and RAF homo- or heterodimerization EC₅₀'s (BRET)⁷. Given the correlations between the compounds binding affinity and RAF dimerization as well as between RAF dimerization and RAS-RAF interaction (revised Fig. 2b), we surmise that binding affinity is a key factor explaining the global effect of inhibitors on RAF complexes. Yet, as shown in our previous work (Lavoie *et al.* (2013) and Thevakumaran *et al.* (2015)), sulfonamide inhibitors, which induce the helix α C-out conformation, are outliers^{7,18}. This implies that the binding mode of RAF inhibitors (α C-in vs out; DFG-in vs out) also plays a key role in dictating differential potencies towards RAF dimerization and RAS-RAF interaction. This was further confirmed in the present manuscript by the fact that paradox breakers, which show equal binding affinity to the RAF kinase domain, have a reduced effect on RAF dimerization and RAS-RAF interaction.

6- What is the effect of the inhibitors on CRAF intramolecular interaction?

This question was also asked by Reviewers #1 (point 2.2) and #2 (point 1). We saw essentially the same response as for BRAF intramolecular interaction. Briefly, CRAF NTR-KD interaction was disrupted by RAF inhibitors, such as GDC-0879 and LY-3009120 (revised Fig. 4b).

7- Supplementary Figures 1c and 2b, effect of PLX on the KRAS^{G12V}-BRAF BRET signal, the EC₅₀ do not match.

These two experiments were independent biological replicates. PLX4032 has a low amplitude (YMAX) effect on the KRAS^{G12V}-BRAF BRET signal, which tends to introduce greater variability for EC₅₀ determination. We conducted two additional and independent dose-response experiments. The Table (shown below) reports the new EC₅₀ determinations, which are consistent with the expected 2 to 3-fold variation for this assay. The average and standard deviation of PLX4032 EC₅₀ and YMAX for the KRAS^{G12V}-BRAF BRET assay are listed in Suppl. Table 1.

	EC ₅₀ determination for PLX4032 on KRAS ^{G12V} - BRAF BRET (nM)
Original Supp. Fig. 1c	117
Original Supp. Fig. 2b	337
Repeat 1	188
Repeat 2	121
Average	191
Stdev	103
SEM	51

While representative single BRET experiments (curves) are shown throughout the Figures, EC₅₀ determinations for each compound reported in the manuscript for the distinct BRET assays are now the means of at least three independent experiments. The data are presented above the specific curves as well as globally reported in Suppl. Fig. 1.

8- *Figure 6, in general, it is problematic to group the inhibitors in this way. Examples are: 1) the pathway activating potential of GDC-0879 is very high, while that of AZ628 and LY3009120 is minimal, yet they are in the same “+++” group. 2) Supplementary Figure 4e shows a very high EC50 for AZ628 (low dimer disruption potential, similar to PLX), yet one compound is in the “+++” group. In addition, it is not clear whether the authors have factored in the widely different compound concentrations needed to achieve the effects at study, and if so, how.*

We agree with the reviewer that grouping the inhibitors the way we did can be problematic given the complex structural and functional outcomes that inhibitors (binding mode and affinity) have on RAF proteins. In the revised Figure (revised Fig. 6c), we removed the Table and schematically presented the position of theoretical compounds along their effects with respect to RAS-RAF association, NTR-KD interaction, and RAF dimerization. We also performed an unbiased hierarchical clustering of the inhibitors profiled in this study based on their BRET EC₅₀'s and YMAX's. This provided an empirical basis to position inhibitor classes along the axis depicted in Fig. 6c.

9- *Discussion, the authors may consider shortening this section.*

We followed the reviewer's advice and shortened the discussion by ~ 20% of its original length by editing various sub-sections.

10- *On page 17, lines 341-42, the authors discuss the possibility that the NTR might “physically occlude the side-to-side interface by interacting with specific residues of the interface”. If this were the case, these residues would not include R509, since the R509H mutation does not impact interaction with NTR (Figure 4e)*

This is correct. The side-to-side interface is relatively large and includes several residues in addition to R509¹⁹. The notion that the NTR might physically occlude this interface therefore does not necessarily require a specific interaction with R509. To remove any ambiguity, we modified the text accordingly (lines 452-456).

11- *On page 20, lines 394-395, the sentence “Concomitantly, these compounds were shown to inhibit kinase domain dimerization as measured in vitro with purified protein” is misleading - the inhibitors promote RAF dimerization in vivo, albeit at a lower level than the type I/II inhibitors.*

We agree with this comment. The sentence was modified as follows (lines 463-465):

“Moreover, these compounds poorly induce BRAF-CRAF heterodimerization in cells and inhibit BRAF kinase domain homodimerization as measured in vitro using purified proteins”

References

- 1 Hatzivassiliou, G. *et al.* RAF inhibitors prime wild-type RAF to activate the MAPK pathway and enhance growth. *Nature* **464**, 431-435 (2010).
- 2 Heidorn, S. J. *et al.* Kinase-dead BRAF and oncogenic RAS cooperate to drive tumor progression through CRAF. *Cell* **140**, 209-221, doi:S0092-8674(09)01626-2 [pii] 10.1016/j.cell.2009.12.040 (2010).
- 3 Poulidakos, P. I., Zhang, C., Bollag, G., Shokat, K. M. & Rosen, N. RAF inhibitors transactivate RAF dimers and ERK signalling in cells with wild-type BRAF. *Nature* **464**, 427-430 (2010).
- 4 Plowman, S. J., Muncke, C., Parton, R. G. & Hancock, J. F. H-ras, K-ras, and inner plasma membrane raft proteins operate in nanoclusters with differential dependence on the actin cytoskeleton. *Proc Natl Acad Sci U S A* **102**, 15500-15505, doi:10.1073/pnas.0504114102 (2005).
- 5 Tian, T. *et al.* Plasma membrane nanoswitches generate high-fidelity Ras signal transduction. *Nat Cell Biol* **9**, 905-914, doi:10.1038/ncb1615 (2007).
- 6 Cho, K. J. *et al.* Raf inhibitors target ras spatiotemporal dynamics. *Curr Biol* **22**, 945-955, doi:10.1016/j.cub.2012.03.067 (2012).
- 7 Lavoie, H. *et al.* Inhibitors that stabilize a closed RAF kinase domain conformation induce dimerization. *Nat Chem Biol* **9**, 428-436, doi:10.1038/nchembio.1257 (2013).
- 8 Charest, P. G., Terrillon, S. & Bouvier, M. Monitoring agonist-promoted conformational changes of beta-arrestin in living cells by intramolecular BRET. *EMBO Rep* **6**, 334-340, doi:10.1038/sj.embor.7400373 (2005).
- 9 Zimmerman, B. *et al.* Differential beta-arrestin-dependent conformational signaling and cellular responses revealed by angiotensin analogs. *Sci Signal* **5**, ra33, doi:10.1126/scisignal.2002522 (2012).
- 10 Chong, H. & Guan, K. L. Regulation of Raf through phosphorylation and N terminus-C terminus interaction. *J Biol Chem* **278**, 36269-36276, doi:10.1074/jbc.M212803200 (2003).
- 11 Cutler, R. E., Jr., Stephens, R. M., Saracino, M. R. & Morrison, D. K. Autoregulation of the Raf-1 serine/threonine kinase. *Proc Natl Acad Sci U S A* **95**, 9214-9219 (1998).
- 12 Tran, N. H., Wu, X. & Frost, J. A. B-Raf and Raf-1 are regulated by distinct autoregulatory mechanisms. *J Biol Chem* **280**, 16244-16253, doi:10.1074/jbc.M501185200 (2005).
- 13 Klemm, J. D. & Pabo, C. O. Oct-1 POU domain-DNA interactions: cooperative binding of isolated subdomains and effects of covalent linkage. *Genes Dev* **10**, 27-36 (1996).

- 14 Kuriyan, J. & Eisenberg, D. The origin of protein interactions and allostery in colocalization. *Nature* **450**, 983-990, doi:10.1038/nature06524 (2007).
- 15 Marais, R., Light, Y., Paterson, H. F., Mason, C. S. & Marshall, C. J. Differential regulation of Raf-1, A-Raf, and B-Raf by oncogenic ras and tyrosine kinases. *J Biol Chem* **272**, 4378-4383 (1997).
- 16 Karoulia, Z. *et al.* An Integrated Model of RAF Inhibitor Action Predicts Inhibitor Activity against Oncogenic BRAF Signaling. *Cancer Cell* **30**, 501-503, doi:10.1016/j.ccell.2016.08.008 (2016).
- 17 Dougherty, M. K. *et al.* Regulation of Raf-1 by direct feedback phosphorylation. *Mol Cell* **17**, 215-224, doi:S1097276504008019 [pii] 10.1016/j.molcel.2004.11.055 (2005).
- 18 Thevakumaran, N. *et al.* Crystal structure of a BRAF kinase domain monomer explains basis for allosteric regulation. *Nat Struct Mol Biol* **22**, 37-43, doi:10.1038/nsmb.2924 (2015).
- 19 Rajakulendran, T., Sahmi, M., Lefrancois, M., Sicheri, F. & Therrien, M. A dimerization-dependent mechanism drives RAF catalytic activation. *Nature* **461**, 542-545 (2009).

REVIEWERS' COMMENTS:

Reviewer #1 (Remarks to the Author):

Overall, the authors have done a mostly adequate job of responding to the reviewers' comments.

Reviewer #2 (Remarks to the Author):

After carefully read this revised manuscript, I think authors have addressed all of my concerns. I have no further questions and strongly recommend it for publishing in Nature Communications.

Reviewer #3 (Remarks to the Author):

The authors have satisfactorily answered my comments. The revised manuscript contains a significant amounts of new information and is in my opinion suitable for publication in Nature Communications.